# DLEFT-MKC: Dynamic Late Fusion Multiple Kernel Clustering with Robust Tensor Learning via Min-Max Optimization

**Yi Zhang**
College of Computer
National University of Defense Technolgy
Changsha, Hunan 410073, China
zhangy@nudt.edu.cn

**Siwei Wang**
Intelligent Game and Decision Lab
Academy of Military Sciences
Beijing 100091, China
wangsiwei13@nudt.edu.cn

**Jiyuan Liu, Shengju Yu, Zhibin Dong, Suyuan Liu, Xinwang Liu*  & En Zhu**
College of Computer, National University of Defense Technolgy
Changsha, Hunan 410073, China
{xinwangliu,enzhu}@nudt.edu.cn

## Abstract

Recent advancements in multiple kernel clustering (MKC) have highlighted the effectiveness of late fusion strategies, particularly in enhancing computational efficiency to near-linear complexity while achieving promising clustering performance. However, existing methods encounter three significant limitations: (1) reliance on fixed base partition matrices that do not adaptively optimize during the clustering process, thereby constraining their performance to the inherent representational capabilities of these matrices; (2) a focus on adjusting kernel weights to explore inter-view consistency and complementarity, which often neglects the intrinsic high-order correlations among views, thereby limiting the extraction of comprehensive multiple kernel information; (3) a lack of adaptive mechanisms to accommodate varying distributions within the data, which limits robustness and generalization. To address these challenges, this paper proposes a novel algorithm termed **D**ynamic **L**at**E** **F**usion Multiple Kernel Clustering with Robust **T**ensor Learning via min-max optimization (**DLEFT-MKC**), which effectively overcomes the representational bottleneck of base partition matrices and facilitates the learning of meaningful high-order cross-view information. Specifically, it is the first to incorporate a min-max optimization paradigm into tensor-based MKC, enhancing algorithm robustness and generalization. Additionally, it dynamically reconstructs decision layers to enhance representation capabilities and subsequently stacks the reconstructed representations for tensor learning that promotes the capture of high-order associations and cluster structures across views, ultimately yielding consensus clustering partitions. To solve the resultant optimization problem, we innovatively design a strategy that combines reduced gradient descent with the alternating direction method of multipliers, ensuring convergence to local optima while maintaining high computational efficiency. Extensive experimental results across various benchmark datasets validate the superior effectiveness and efficiency of the proposed DLEFT-MKC.

## 1 Introduction

Multiple Kernel Clustering (MKC) has emerged as a crucial technique in machine learning, aimed at analyzing complex linearly-inseparable data by projecting data features into higher-dimensional or even infinite-dimensional spaces, Reproducing Kernel Hilbert Space (RKHS), thus transforming data into linearly separable entities (Filippone et al., 2008; Marin et al., 2017; Blanco Valencia et al., 2017). Given today's era of big data, where almost all data encompass multiple distinct representations or views, MKC algorithms have naturally garnered considerable attention and study in the field. They integrate multi-source information within the kernel space, subsequently assigning samples to distinct clusters (Gönen & Alpaydın, 2011; Kumar & Daumé, 2011; Chitta et al.,

---

*corresponding author

2012; Tang et al., 2022).Specifically, MKC primarily learns an optimally combined kernel by mining information from multiple views, subsequently serving the clustering tasks. This methodology is particularly beneficial across various real-world applications, including image recognition, natural language processing, anomaly detection, and bioinformatics (Peng et al., 2019; Zhou et al., 2020; Zhang et al., 2022a; Wang et al., 2022; Yu et al., 2023b;a; Tang et al., 2023; Yu et al., 2024).

Some representative MKC algorithms believe that the optimal kernel is a linear combination of base kernels (Huang et al., 2012; Gönen & Margolin, 2014; Bang et al., 2018). The neighborhood kernel learning methods seek a non-linear combination of base kernels for better representability of the optimal kernel (Liu et al., 2017; 2020). A matrix-induced regularization is introduced to consider the selection of kernels (Liu et al., 2016; Hu et al., 2019). Recently, a min-max framework has been introduced to seek optimism in pessimism (Liu, 2023b), and some variants are proposed (Liu et al., 2021c; Liu, 2023a). An overall process fusion manner is proposed to deepen the degree of fusion between views (Zhang et al., 2022b). Furthermore, tensor-based multi-view clustering algorithms also received a lot of attention due to their capability to study the high-order correlations among views and achieve encouraging performance. Wu et al. (2019) reorganize the affinity matrices into tensor form and learn its intrinsic tensor based on low-rank tensor approximation. The weighted t-TNN is introduced to reflect the importance of different eigenvalues (Gao et al., 2020), A new graph learning paradigm is proposed to enable the affinity graph propagated from KKM to enjoy the valuable block diagonal and sparse property through an explicit theoretical connection between the clustering indicator matrix and affinity graph (Ren et al., 2021). Chen et al. (2022c) stacks multiple affinity representations in a low-rank constrained tensor to recover their comprehensiveness and higher-order correlations. Late fusion MKC methods propose to first cluster each individual view and then fuse these results into a cohesive solution (Wang et al., 2019b). The development of late fusion strategies has further transformed MKC techniques (Zhang et al., 2021; Liu et al., 2021b); these approaches not only benefit exploring cluster structures during the fusion process but also significantly enhance computational efficiency, achieving near-linear complexity.

Despite the encouraging improvement in clustering performance, several critical challenges remain unaddressed in MKC. First, late fusion MKC relies on fixed initial base partitions that do not adaptively optimize during the clustering process, presenting a bottleneck in performance due to their inherent representational limitations; suboptimal starting points can severely compromise final outcomes. Additionally, many existing MKC algorithms focus on adjusting kernel weights to explore inter-view consistency and complementarity, and often overlook the intrinsic high-order correlations among views, thereby limiting the extraction of comprehensive multiple kernel information. Furthermore, existing MKC methods frequently lack adaptive mechanisms that accommodate varying distributions within the data, thus limiting their robustness and generalization.

To tackle these issues, we propose a novel algorithm termed **D**ynamic **L**at**E** **F**usion Multiple Kernel Clustering with Robust **T**ensor Learning via Min-Max Optimization (**DLEFT-MKC**). Specifically, we, for the first time, incorporate a min-max optimization paradigm into tensor-based MKC, representing a pioneering exploration aimed at enhancing both performance robustness and generalization in clustering. Additionally, DLEFT-MKC dynamically reconstructs and calibrates the base partitions, effectively overcoming the representational bottleneck of initial base partitions. Furthermore, stacking the dynamically adjusted partition matrices into tensors while applying t-TNN constraints promotes the learning of meaningful higher-order correlations and cluster structures across views. We design an innovative and efficient strategy that combines the reduced gradient descent method (RGDM) with the alternating direction method of multipliers (ADMM) to solve the resultant max-min-max optimization problem, ensuring convergence to local optima while maintaining high computational efficiency. For evaluating the proposed algorithm, we conduct comprehensive experimental studies in terms of clustering performance, evolution and convergence, cluster partitions, parameter sensitivity, ablation study, and time complexity. Extensive experimental results across various benchmark datasets validate the effectiveness and efficiency of our proposed DLEFT-MKC.

The primary contributions of this paper are summarized as follows,

- This study is the first to incorporate a min-max optimization paradigm into tensor-based MKC, which represents a pioneering exploration of min-max optimization aimed at enhancing both performance and robustness in clustering.
- We propose a groundbreaking approach for the dynamical reconstruction and calibration of base partition matrices from LFMVC, effectively overcoming their representational bottleneck and enhancing clustering performance.

- We stack the reconstructed representations into tensors and optimize dynamic partitions using tensor techniques, significantly enhancing our ability to learn high-order correlations and uncover latent structures across views.
- To solve the resultant optimization problem, We design an innovative and efficient strategy to combine the RGDM with ADMM. Extensive experimental results across various benchmark datasets validate both the effectiveness and efficiency of our proposed algorithm.

## 2 RELATED WORK

### 2.1 MULTIPLE KERNEL $k$-MEANS CLUSTERING (MKKM)

The $k$-means clustering algorithm aims to partition data points into $k$ clusters by minimizing intra-cluster distances and maximizing inter-cluster distances. Its objective can be articulated as follows:

$$\min_{\boldsymbol{S}, \boldsymbol{c}} \sum\nolimits_{i=1}^{n} \sum\nolimits_{j=1}^{k} S_{ij} \|\boldsymbol{x}_i - \boldsymbol{c}_j\|^2, \ s.t. \ \boldsymbol{S}\boldsymbol{1} = \boldsymbol{1}, \tag{1}$$

where $\boldsymbol{x}_i$ represents the $i$-th data sample, $\boldsymbol{c}_j$ denotes the center of the $j$-th cluster, $\boldsymbol{S} \in \mathbb{R}^{n \times k}$ serves as the clustering assignment matrix. If the $i$-th sample is assigned to the $j$-th cluster, then $\boldsymbol{S}_{ij} = 1$. and $n$ and $k$ denote the number of samples and clusters, respectively.

Many approaches capture structural information by mapping features into Reproducing Kernel Hilbert Space (RKHS) to address complex data that is linearly inseparable. Notably, the dimensionality of mapped features can be extremely high or even infinite; therefore, kernel methods are typically employed to compute the kernel matrix, thereby avoiding explicit mapping. By defining $\boldsymbol{F} = \boldsymbol{S}\boldsymbol{L}^{(\frac{1}{2})}$ the clustering assignment matrix $\boldsymbol{S}$ is relaxed into the real domain, where $\boldsymbol{L} \in \mathbb{R}^{k \times k}$ is a diagonal matrix with each diagonal element representing the reciprocal of the sum of its corresponding column in matrix $\boldsymbol{S}$. Consequently, the kernel K-means clustering algorithm can be articulated as follows:

$$\min_{\boldsymbol{F}} \ \mathrm{Tr}\left(\boldsymbol{K}(\boldsymbol{I} - \boldsymbol{F}\boldsymbol{F}^{\top})\right), \ s.t. \ \boldsymbol{F}^{\top}\boldsymbol{F} = \boldsymbol{I}, \tag{2}$$

where $\boldsymbol{K}$ denotes the kernel matrix calculated using an implicit mapping function $\phi(\cdot)$.

Following the framework of multiple kernel learning (Rakotomamonjy et al., 2008), the kernel K-means method can be extended to multi-view scenarios, assuming that an optimal consensus kernel matrix can be derived as a linear combination of predefined base kernel matrices. Therefore, the framework of multiple kernel K-means clustering can be formally articulated as follows:

$$\min_{\boldsymbol{F}, \boldsymbol{\gamma}} \mathrm{Tr}\left(\boldsymbol{K}_{\boldsymbol{\gamma}}(\boldsymbol{I} - \boldsymbol{F}\boldsymbol{F}^{\top})\right), \ s.t. \ \boldsymbol{F}^{\top}\boldsymbol{F} = \boldsymbol{I}, \ \boldsymbol{\gamma} \in \Delta, \tag{3}$$

where $\boldsymbol{K}_{\boldsymbol{\gamma}} = \sum_{p=1}^{m} \gamma^2 \boldsymbol{K}_p$ denotes a combination of kernel matrices from different views, $\boldsymbol{K}_p$ is the kernel matrix of $p$-th view, $\boldsymbol{\gamma}_p$ serve as the corresponding weight coefficient for each kernel view with $\Delta = \{\boldsymbol{\gamma} \in \mathbb{R}^m | \sum_{p=1}^{m} \boldsymbol{\gamma}_p = 1, \gamma_q \geq 0, \forall p\}$, $m$ denotes the number of views. According to the existing literature, the optimization problem of MKKM can typically be solved using coordinate descent optimization techniques that iteratively optimize specific variables while keeping others fixed.

### 2.2 LATE FUSION MULTI-VIEW CLUSTERING

Recently, the literature(Wang et al., 2019b) has proposed a method known as Late Fusion Multi-view Clustering (LFMVC) to address the computational complexity challenges associated with MKC. Unlike traditional multiple kernel K-means methods that represent distribution information from different views through a weighted combination of kernel matrices $\{\boldsymbol{K}_p\}_{p=1}^{m} \in \mathbb{R}^{n \times n}$, LFMVC integrates information at the decision level by utilizing smaller base partition matrices $\{\boldsymbol{F}_p\}_{p=1}^{m} \in \mathbb{R}^{n \times k}$ to capture data distributions for each kernel view. This strategy significantly reduces both time and memory overhead during the MKC process. Specifically, Late fusion MKC aims to learn a consensus clustering partition matrix $\boldsymbol{F}^* \in \mathbb{R}^{n \times k}$ by integrating individual base partition matrices $\{\boldsymbol{F}_p\}_{p=1}^{m}$. Its objective function emphasizes maximizing alignment between weighted base partitions generated from different views and the consensus partition:

$$\max_{\boldsymbol{F}, \boldsymbol{T}_p, \boldsymbol{\gamma}} \ \mathrm{Tr}\left(\boldsymbol{F}^{\top} \sum\nolimits_{p=1}^{m} \gamma_p \boldsymbol{F}_p \boldsymbol{T}_p\right), \ s.t. \ \boldsymbol{F}^{\top}\boldsymbol{F} = \boldsymbol{I}, \boldsymbol{T}_p^{\top}\boldsymbol{T}_p = \boldsymbol{I}, \ \forall p, \boldsymbol{\gamma} \in \nabla, \tag{4}$$

where $\boldsymbol{\gamma}$ denotes the weight coefficients of various kernel views, $\nabla = \{\boldsymbol{\gamma} \in \mathbb{R}^m \mid \sum_{p=1}^{m} \gamma_p^2 = 1, \ \gamma_p \geq 0, \ \forall p\}$, and $\boldsymbol{T}_p \in \mathbb{R}^{k \times k}$ is the $p$-th permutation matrix, for the better alignment among base partitions from various views.

We can observe from Eq.(4) that LFMVC aims to optimize its objective function by maximizing all involved variables. To achieve this objective, a coordinate descent method has been developed for optimization purposes. As analyzed in previous studies (Wang et al., 2019b), LFMVC's near-linear computational complexity and efficiency enable it to handle large-scale clustering tasks effectively.

## 2.3 PRELIMINARIES OF 3-ORDER TENSOR

### 2.3.1 TENSOR SINGULAR VALUE DECOMPOSITION (T-SVD)

For a tensor $\mathbf{A} \in \mathbb{R}^{n_1 \times n_2 \times n_3}$, its t-SVD can be factorized as $\mathbf{A} = \mathbf{U} * \mathbf{S} * \mathbf{V}^\top$, where $\mathbf{U} \in \mathbb{R}^{n_1 \times n_1 \times n_3}$ and $\mathbf{S} \in \mathbb{R}^{n_1 \times n_2 \times n_3}$ are orthogonal tensors, and $\mathbf{V} \in \mathbb{R}^{n_1 \times n_2 \times n_3}$ is an f-diagonal tensor, whose each frontal slices is a diagonal matrix. According to the literature (Kilmer et al., 2013; Kilmer & Martin, 2011), the above t-SVD problem can be efficiently settled by matrix SVD in the Fourier domain, i.e., $\overline{\boldsymbol{A}}_k = \overline{\boldsymbol{U}}_k \overline{\boldsymbol{S}}_k \overline{\boldsymbol{V}}_k^\top, k = 1, 2, \cdots, n_3$.

### 2.3.2 T-SVD BASED TENSOR NUCLEAR NORM (T-TNN)

For a tensor $\mathbf{A} \in \mathbb{R}^{n_1 \times n_2 \times n_3}$, its t-TNN can be expressed as, $||\mathbf{A}||_{\circledast} = \sum_{k=1}^{n_3} ||\overline{\boldsymbol{A}}_k||_* = \sum_{k=1}^{n_3} \sum_{i=1}^{min(n_1,n_2)} \sigma_i(\overline{\boldsymbol{A}}_k)$, where $\sigma_i(\overline{\boldsymbol{A}}_k)$ denotes the $i$-th largest singular value of $\overline{\boldsymbol{A}}_k$.

Note that according to (Zhang et al., 2014; Semerci et al., 2014), t-TNN is proven to be valid and the tightest convex relaxation to $l_1$-norm of the tensor multi-rank.

# 3 PROPOSED

## 3.1 FORMULATION

We propose a novel dynamic late-fusion multiple kernel clustering algorithm based on robust tensor learning through min-max optimization, effectively addressing the representational bottleneck of base partition matrices and facilitating the acquisition of meaningful high-order cross-view information. Specifically, we first incorporate a min-max optimization paradigm into tensor-based MKC, which represents a pioneering exploration of min-max optimization designed to enhance both performance and robustness in clustering. Additionally, the proposed algorithm dynamically reconstructs and calibrates the base partition matrix, effectively overcoming constraints imposed by initial representational limitations. Furthermore, stacking the dynamically adjusted partition matrices into tensors while applying t-TNN constraints promotes the learning of higher-order correlations and cluster structures across views.

To do so, we first introduce the dynamic partitions $\{\hat{\boldsymbol{F}}_p\}_{p=1}^m$ to reconstruct the base partition matrices $\{\boldsymbol{F}_p\}_{p=1}^m$ of late fusion strategy based MKC. Next, We maximize the alignment between the reconstructed and base partitions to ensure the quality of reconstruction, and dynamically optimize this alignment during the subsequent iterations. Furthermore, to explore and capture higher-order intrinsic correlations across views, we stack the dynamic reconstruction $\{\hat{\boldsymbol{F}}_p\}_{p=1}^m$ into a tensor $\hat{\mathbf{F}}$ and optimize it with t-TNN. Additionally, we impose an orthogonal constraint on it to preserve its capacity to reveal the clustering structure. Thus we can obtain the following expression:

$$\max_{\hat{\mathbf{F}}, \hat{\boldsymbol{F}}_p} \sum_{p=1}^m \text{Tr}(\hat{\boldsymbol{F}}_p^\top \boldsymbol{F}_p) - \rho ||\hat{\mathbf{F}}||_{\circledast}, \ s.t. \ \hat{\boldsymbol{F}}_p^\top \hat{\boldsymbol{F}}_p = \boldsymbol{I}, \forall p. \tag{5}$$

Next, we attempt to directly learn the consensus clustering partition by incorporating Eq.(5) and permutation matrices $\{\boldsymbol{T}_p\}_{p=1}^m$ with the paradigm of LFMVC. In addition, due to the different contributions of various views, we assign kernel weight coefficients $\gamma$ to each view in order to sufficiently mine and learn each kernel view with particular emphasis.

Finally, we pioneeringly introduce the min-max paradigm into the resultant objective function, which minimizes the function w.r.t. $\gamma$ and maximizes it w.r.t. $\hat{\mathbf{F}}, \boldsymbol{F}^*, \hat{\boldsymbol{F}}_p$ and $\boldsymbol{T}_p$. Therefore, the final objective function can be expressed as follows,

$$\max_{\hat{\mathbf{F}}, \hat{\boldsymbol{F}}_p, \boldsymbol{T}_p} \min_{\gamma} \max_{\boldsymbol{F}^*} \text{Tr}(\boldsymbol{F}^{*\top}(\sum_{p=1}^m \gamma_p^2 \hat{\boldsymbol{F}}_p \boldsymbol{T}_p)) + \lambda \sum_{p=1}^m \gamma_p^2 \text{Tr}(\hat{\boldsymbol{F}}_p^\top \boldsymbol{F}_p) - \rho ||\hat{\mathbf{F}}||_{\circledast},$$

$$s.t. \ \hat{\boldsymbol{F}}_p^\top \hat{\boldsymbol{F}}_p = \boldsymbol{I}, \boldsymbol{T}_p^\top \boldsymbol{T}_p = \boldsymbol{I}, \forall p, \gamma \in \Delta, \boldsymbol{F}^{*\top} \boldsymbol{F}^* = \boldsymbol{I}. \tag{6}$$

This max-min-max paradigm denotes that we maximize the alignment between the consensus clustering partition and base partitions while optimizing the kernel weight coefficients to minimize the

objective function, preventing premature convergence to local optima. In this way, the proposed algorithm can robustly learn the optimal consensus clustering partition even under challenging conditions.

## 3.2 OPTIMIZATION

To solve the resultant max-min-max optimization problem of DLEFT-MKC in Eq.(6), we combine the optimization strategies of the reduced gradient descent method (RGDM) and Alternating Direction Method of Multipliers (ADMM), updating one specific variable while keeping others fixed. To facilitate the divisibility of $\mathbf{F}$, we introduce an auxiliary tensor variable $\mathbf{A}$ according to the principles of ADMM and obtain the augmented Lagrangian function as follows,

$$\mathcal{L}(\mathbf{A}, \hat{\boldsymbol{F}}_p, \boldsymbol{T}_p, \boldsymbol{\gamma}, \boldsymbol{F}^*) = \sum_{p=1}^{m} \gamma_p^2 \mathrm{Tr}(\boldsymbol{F}^{*\top} \hat{\boldsymbol{F}}_p \boldsymbol{T}_p + \lambda \hat{\boldsymbol{F}}_p^\top \boldsymbol{F}_p) - \rho ||\mathbf{A}||_\circledast - \frac{\mu}{2} ||\mathbf{A} - (\hat{\mathbf{F}} + \frac{\mathbf{Y}}{\mu})||_F^2, \quad (7)$$

$$s.t. \ \hat{\boldsymbol{F}}_p^\top \hat{\boldsymbol{F}}_p = \boldsymbol{I}, \boldsymbol{T}_p^\top \boldsymbol{T}_p = \boldsymbol{I}, \forall p, \boldsymbol{\gamma} \in \Delta, \boldsymbol{F}^{*\top} \boldsymbol{F}^* = \boldsymbol{I},$$

where, $\mathbf{Y} \in \mathbb{R}^{n \times k \times m}$ represents the Lagrange multiplier, with $\mu > 0$ acting as the penalization factor. An alternating optimization strategy allows for the decomposition of the problem in Eq.(7) into five distinct sub-problems. Each sub-problem independently optimizes its respective variables while keeping others fixed.

**update $\{\hat{\boldsymbol{F}}_p\}_{p=1}^m$:** By fixing the other variables, $\{\hat{\boldsymbol{F}}_p\}_{p=1}^m$ can be updated as follows,

$$\max_{\hat{\boldsymbol{F}}_p} \sum_{p=1}^m \gamma_p^2 \mathrm{Tr}(\boldsymbol{F}^{*\top} \hat{\boldsymbol{F}}_p \boldsymbol{T}_p + \lambda \hat{\boldsymbol{F}}_p^\top \boldsymbol{F}_p) - \frac{\mu}{2} ||\mathbf{A} - (\hat{\mathbf{F}} + \frac{\mathbf{Y}}{\mu})||_F^2, \ s.t. \ \hat{\boldsymbol{F}}_p^\top \hat{\boldsymbol{F}}_p = \boldsymbol{I}. \quad (8)$$

Then by expanding the Frobenius norm and simplifying this problem, we can obtain the following problem w.r.t. each $\hat{\boldsymbol{F}}_p$:

$$\max_{\hat{\boldsymbol{F}}_p} \mathrm{Tr}(\hat{\boldsymbol{F}}_p^\top (\gamma_p^2 \boldsymbol{F}^* \boldsymbol{T}_p^\top + \lambda \gamma_p^2 \boldsymbol{F}_p + \mu \boldsymbol{A}_p - \boldsymbol{Y}_p)), \ s.t. \ \hat{\boldsymbol{F}}_p^\top \hat{\boldsymbol{F}}_p = \boldsymbol{I}, \quad (9)$$

where $\boldsymbol{A}_p$ and $\boldsymbol{Y}_p$ represent the $p$-th slice of $\mathbf{A}$ and $\mathbf{Y}$, respectively. By setting $\boldsymbol{M}_p = \gamma_p^2 \boldsymbol{F}^* \boldsymbol{T}_p^\top + \lambda \gamma_p^2 \boldsymbol{F}_p + \mu \boldsymbol{A}_p - \boldsymbol{Y}_p$, the problem in Eq.(9) can be effectively addressed by applying the economic rank-$k$ SVD of $\boldsymbol{M}_p$. Assume that the matrix $\boldsymbol{M}_p$ possesses a rank-$k$ truncated SVD representation given by $\boldsymbol{M}_p = \boldsymbol{U}_k \Sigma_k \boldsymbol{V}_k^\top$, where $\boldsymbol{U}_k \in \mathbb{R}^{n \times k}, \Sigma_k \in \mathbb{R}^{k \times k}, \boldsymbol{V}_k \in \mathbb{R}^{k \times k}$. Then, the problem in Eq.(9) has a closed-form optimal solution given by,

$$\hat{\boldsymbol{F}}_p = \boldsymbol{U}_k \boldsymbol{V}_k^\top. \quad (10)$$

**update $\boldsymbol{\gamma}$ and $\boldsymbol{F}^*$:** By fixing the other variables, we derive a min-max optimization problem w.r.t. $\boldsymbol{\gamma}$ and $\boldsymbol{F}^*$ as follows,

$$\min_{\boldsymbol{\gamma}} \max_{\boldsymbol{F}^*} \sum_{p=1}^m \gamma_p^2 \mathrm{Tr}(\boldsymbol{F}^{*\top} \hat{\boldsymbol{F}}_p \boldsymbol{T}_p + \lambda \hat{\boldsymbol{F}}_p^\top \boldsymbol{F}_p), \ s.t. \ \boldsymbol{\gamma} \in \Delta, \boldsymbol{F}^{*\top} \boldsymbol{F}^* = \boldsymbol{I}. \quad (11)$$

To solve it, we begin by rewriting it as an optimal value function of the maximization problem as follows,

$$\min_{\boldsymbol{\gamma} \in \Delta} \mathcal{G}(\boldsymbol{\gamma}), \quad \mathcal{G}(\boldsymbol{\gamma}) = \left\{ \max_{\boldsymbol{F}^*} \mathrm{Tr}(\boldsymbol{F}^{*\top} (\sum_{p=1}^m \gamma_p^2 \hat{\boldsymbol{F}}_p \boldsymbol{T}_p) + \lambda \sum_{p=1}^m \gamma_p^2 \hat{\boldsymbol{F}}_p^\top \boldsymbol{F}_p) \right\}. \quad (12)$$

According to Theorem 4.1 in the literature (Bonnans & Shapiro, 1998), the optimal value funtion $\mathcal{G}(\boldsymbol{\gamma})$ in Eq.(12) is differentiable, and $\frac{\partial \mathcal{G}(\boldsymbol{\gamma})}{\partial \gamma_p} = 2\gamma_p \mathrm{Tr}\left(\overline{\boldsymbol{F}^*}^\top \hat{\boldsymbol{F}}_p \boldsymbol{T}_p + \lambda \hat{\boldsymbol{F}}_p^\top \boldsymbol{F}_p\right)$, where $\overline{\boldsymbol{F}^*} = \left\{ \arg\max_{\boldsymbol{F} \in \Gamma} \mathrm{Tr}\left(\boldsymbol{F}^* \left(\sum_{p=1}^m \gamma_p^2 \hat{\boldsymbol{F}}_p \boldsymbol{T}_p\right)\right) \right\}$. Therefore, a reduced gradient descent strategy can be employed to address the optimization problem in Eq.(12). According to the literature (Liu, 2023b), we firstly calculate the gradient of $\mathcal{G}(\boldsymbol{\gamma})$ w.r.t. $\boldsymbol{\gamma}$, and subsequently optimize $\boldsymbol{\gamma}$ along a descent direction while satisfying the constraint $\boldsymbol{\gamma} \in \Delta$, with the optimal $\boldsymbol{F}^*$.

To do so, we should guarantee the equality constraint and positivity constraint of $\boldsymbol{\gamma}$. First, we designate $\gamma_u$ as a non-zero component of $\boldsymbol{\gamma}$ and $\triangledown\mathcal{G}(\boldsymbol{\gamma})$ as the reduced gradient of $\mathcal{G}(\boldsymbol{\gamma})$. By following Rakotomamonjy et al. (2008); Liu (2023b), $\triangledown\mathcal{G}(\boldsymbol{\gamma})$ can be expressed as follows,

$$[\triangledown\mathcal{G}(\boldsymbol{\gamma})]_p = \frac{\partial \mathcal{G}(\boldsymbol{\gamma})}{\partial \gamma_p} - \frac{\partial \mathcal{G}(\boldsymbol{\gamma})}{\partial \gamma_u} \ \forall p \neq u, [\triangledown\mathcal{G}(\boldsymbol{\gamma})]_u = \sum_{p=1, p \neq u}^m \left(\frac{\partial \mathcal{G}(\boldsymbol{\gamma})}{\partial \gamma_u} - \frac{\partial \mathcal{G}(\boldsymbol{\gamma})}{\partial \gamma_p}\right), \quad (13)$$

where $u$ typically denotes the index of the largest component of $\boldsymbol{\gamma}$, as suggested by Rakotomamonjy et al. (2008), which is regarded as providing improved numerical stability.

Next, to ensure that $\boldsymbol{\gamma}$ remains positive at all times, we design the calculation strategy of the descent direction for updating $\boldsymbol{\gamma}$ as follows,

$$v_p = \begin{cases} 0 & \text{if } \gamma_p = 0 \text{ and } [\triangledown \mathcal{G}(\boldsymbol{\gamma})]_p > 0, \\ -[\triangledown \mathcal{G}(\boldsymbol{\gamma})]_u & \text{if } p = u, \\ -[\triangledown \mathcal{G}(\boldsymbol{\gamma})]_p & \text{otherwise.} \end{cases} \tag{14}$$

After deriving the descent direction $\boldsymbol{V} = [v_1, \cdots, v_m]^\top$ from Eq.(14), we can then update the weights $\boldsymbol{\gamma}$ using $\boldsymbol{\gamma} \leftarrow \boldsymbol{\gamma} + \alpha \boldsymbol{V}$, where $\alpha$ is a chosen step size that could be determined using line search strategies such as Armijo's rule. Overall, the algorithm for solving the optimization problem in Eq.(12) is outlined in Algorithm 1 in the appendix.

**update A:** By fixing the other variables, the **A** sub-problem constitutes a t-TNN minimization problem and can be articulated as follows,

$$\min_{\mathbf{A}} \rho ||\mathbf{A}||_\circledast + \frac{\mu}{2} ||\mathbf{A} - (\hat{\mathbf{F}} + \frac{\mathbf{Y}}{\mu})||_F^2. \tag{15}$$

Let $\mathbf{B} = \hat{\mathbf{F}} + \frac{\mathbf{Y}}{\mu}$, the sub-problem 15 can be addressed using the tensor tubal-shrinkage of $\mathbf{B}$, as detailed in Theorem 1.

**Theorem 1** (Zhou et al., 2019a) Given $\mathbf{A}, \mathbf{B} \in \mathbb{R}^{n_1 \times n_2 \times n_3}$, $l = min(n1, n2)$, we can have $\mathbf{A} = \mathbf{U} * \mathbf{S} * \mathbf{V}^\top$ by t-SVD. The global optimal solution to $\min_{\mathbf{A}} \rho ||\mathbf{A}||_\circledast + \frac{1}{2} ||\mathbf{A} - \mathbf{B}||_F^2$ is provided by the tensor tubal-shrinkage operator, i.e., $\mathbf{A} = \Gamma_\tau(\mathbf{B}) = \mathbf{U} * ifft(\mathbf{P}_\tau(\overline{\mathbf{B}})) * \mathbf{V}^\top$, where $\overline{\mathbf{B}} = fft(\mathbf{B}, [], 3)$, and $\mathbf{P}_\tau(\overline{\mathbf{B}})$ is a tensor whose $i$-th frontal slice is $\mathbf{P}_\tau(\overline{\mathbf{B}}_i) = diag(\xi_1, \xi_2, \cdots, \xi_l)$ with $\xi_i = sign(\sigma_i(\overline{\mathbf{B}}_i))max(\sigma_i(\overline{\mathbf{B}}_i) - \tau, 0)$.

**update $\{\boldsymbol{T}_p\}_{p=1}^m$:** By fixing the other variables, $\{\boldsymbol{T}_p\}_{p=1}^m$ sub-problems can be addressed as follows,

$$\max_{\boldsymbol{T}_p} \sum_{p=1}^m \gamma_p^2 \text{Tr}(\boldsymbol{F}^{*\top} \hat{\boldsymbol{F}}_p \boldsymbol{T}_p) \ s.t. \ \hat{\boldsymbol{T}}_p^\top \hat{\boldsymbol{T}}_p = \boldsymbol{T}_p^\top \boldsymbol{T}_p = \boldsymbol{I}, \tag{16}$$

which can be readily solved in a manner similar to that of the problem in Eq.(9).

**update Y and $\mu$:** The penalty factor $\mu$ and the Lagrange multiplier **Y** are updated as follows,
$$\mathbf{Y} = \mathbf{Y} + \mu(\mathbf{F} - \mathbf{A}), \mu = \tau \times \mu, \tag{17}$$
where the literature typically sets $\tau > 1$ to enhance convergence speed (Chen et al., 2020), and we set $\tau = 2$ in this paper.

In conclusion, we present the algorithm process of DLEFT-MKC in Algorithm 2 in the appendix.

## 4 EXPERIMENT

### 4.1 EXPERIMENT SETTING

We utilize multiple benchmark datasets to evaluate the clustering performance of DLEFT-MKC, including: *Liver*[1] *BBCSport*[2], *ProteinFold*[3], *Willow*[4], *Plant*[5], *PsortNeg*[5], *Scene15* (Lazebnik et al., 2006), *CCV*[6], *Flower102*[7], *Reuters*[8]. Tab. 1 summarizes the detailed information regarding the utilized datasets. These datasets exhibit considerable variation in sample sizes (345 to 18,758), kernel counts (2 to 69), and cluster numbers (2 to 102), thereby offering a balanced experimental platform for evaluating the clustering performance of differing algorithms. For all datasets, the true number of clusters $k$ is predetermined and provided

| Dataset | #Samples | #kernel | #clusters |
|---|---|---|---|
| Liver | 345 | 6 | 2 |
| Bbcsport | 544 | 2 | 5 |
| ProteinFold | 694 | 12 | 27 |
| Willow | 911 | 3 | 7 |
| Plant | 940 | 69 | 4 |
| PsortNeg | 1444 | 69 | 5 |
| Scene15 | 4485 | 3 | 15 |
| CCV | 6773 | 3 | 20 |
| Flower102 | 8189 | 4 | 102 |
| Reuters | 18758 | 5 | 6 |

Table 1: Summary of datasets used.

---

[1] https://archive.ics.uci.edu/dataset/

[2] http://mlg.ucd.ie/datasets/

[3] http://mkl.ucsd.edu/dataset/protein-fold-prediction

[4] https://github.com/wangsiwei2010/awesome-multi-view-clustering

[5] https://bmi.inf.ethz.ch/supplements/protsubloc/

[6] www.ee.columbia.edu/ln/dvmm/CCV/

[7] https://www.robots.ox.ac.uk/~vgg/data/flowers/

[8] https://kdd.ics.uci.edu/databases/reuters21578/

as input. We apply four commonly used criteria: clustering accuracy (ACC), normalized mutual information (NMI), purity (PUR), and rand index (RI). We evaluate the proposed DLEFT-MKC in terms of clustering performance, evolution and convergence, cluster partitions, parameter sensitivity, ablation study, and time complexity. The complete experimental results, along with sufficient instructions for reproducibility, are provided in the appendix.

Along with **DLEFT-MKC**, we compare it against numerous MKC algorithms selected from recent literature. Specifically, **Avg-KKM** and **SB-KKM** serve as baselines that perform KKM on average kernel and single kernel without additional operations. We also include classical MKC algorithms such as **MKKM** (Huang et al., 2012), **LMKKM** (Gönen & Margolin, 2014), **ONKC** (Liu et al., 2017), **MKKM-MR** (Liu et al., 2016), and **LKAM** (Li et al., 2016). Furthermore, we incorporate some recent methods including subspace-based, graph-based, and tensor-based approaches, i.e., **LFMVC** (Wang et al., 2019b), **NKSS** (Zhou et al., 2019b), **SPMKC** (Ren & Sun, 2020), **HMKC** (Liu et al., 2021a), **SMKKM** (Liu, 2023b), **OPLFMVC** (Liu et al., 2021b), **LSMKKM** (Liu et al., 2021c), **AIMC** (Chen et al., 2022a), **OMSC** (Chen et al., 2022b), **HFLSMKKM** (Liu, 2023a), **GMC** (Wang et al., 2019a), **LTBPL** (Chen et al., 2022c), **UGLTL** (Wu et al., 2019), **WTNNM** (Gao et al., 2020), **KCGT** (Ren et al., 2021).

For all algorithms, we adhere to guidelines in the literature for parameter configuration. In addition, each experiment is conducted 20 times using $k$-means to reduce the adverse impact of randomness. The average results, along with standard deviations, are then reported.

## 4.2 Experimental Results

### 4.2.1 Clustering Performance

Tab.2 presents a comparison of ACC among the aforementioned algorithms, where '-' signifies that the results are unavailable due to an out-of-memory error, with the top three results being highlighted. Note that comparisons regarding NMI, PUR, and RI are included in the appendix. Then the following observations can be drawn:

(1) Recent advancements in clustering algorithms have demonstrated that tensor-based clustering methods yield significant performance improvements, particularly when compared to traditional MKC algorithms. For instance, the algorithms LTBPL, UGLTL, WTNNM, and KCGT, which are based on tensor learning, consistently outperformed conventional methods like SMKKM and HMKC across all ten datasets. Specifically, UGLTL and WTNNM achieved average ACC improvements of approximately $14.8\%$ and $7.8\%$, respectively, over the best-performing traditional MKC.

(2) Despite these advancements, these tensor-based approaches still face challenges related to computational efficiency and scalability. For example, LTBPL, UGLTL, and WTNNM are unable to handle the Reuters dataset effectively. In contrast, our proposed DLEFT-MKC demonstrates a sig-

| Algorithms | Liver | BBCSport | ProteinFold | Willow | Plant | PsortNeg | Scene15 | CCV | Flower102 | Reuters |
|---|---|---|---|---|---|---|---|---|---|---|
| Avg-KKM | $54.2_{\pm 0.0}$ | $63.2_{\pm 1.4}$ | $29.0_{\pm 1.5}$ | $22.2_{\pm 0.3}$ | $61.3_{\pm 0.9}$ | $41.0_{\pm 1.4}$ | $43.2_{\pm 1.8}$ | $19.6_{\pm 0.6}$ | $27.1_{\pm 0.8}$ | $45.5_{\pm 1.5}$ |
| SB-KKM | $57.9_{\pm 0.1}$ | $71.4_{\pm 0.1}$ | $33.8_{\pm 1.3}$ | $26.8_{\pm 0.3}$ | $51.2_{\pm 1.1}$ | $55.3_{\pm 0.0}$ | $39.3_{\pm 0.2}$ | $20.1_{\pm 0.2}$ | $33.0_{\pm 1.0}$ | $47.2_{\pm 0.0}$ |
| MKKM | $55.0_{\pm 0.3}$ | $63.0_{\pm 1.5}$ | $27.0_{\pm 1.1}$ | $22.0_{\pm 0.2}$ | $56.1_{\pm 0.6}$ | $51.9_{\pm 0.3}$ | $41.2_{\pm 0.1}$ | $18.0_{\pm 0.5}$ | $22.4_{\pm 0.5}$ | $45.4_{\pm 1.5}$ |
| LMKKM | $53.7_{\pm 1.1}$ | $63.9_{\pm 1.4}$ | $22.4_{\pm 0.7}$ | $22.6_{\pm 0.2}$ | - | - | $40.9_{\pm 0.1}$ | $18.6_{\pm 0.1}$ | - | - |
| ONKC | $52.9_{\pm 1.9}$ | $63.4_{\pm 1.4}$ | $36.3_{\pm 1.5}$ | $22.6_{\pm 0.4}$ | $41.4_{\pm 0.2}$ | $40.2_{\pm 0.6}$ | $39.9_{\pm 1.4}$ | $22.4_{\pm 0.3}$ | $39.5_{\pm 0.7}$ | $41.8_{\pm 1.2}$ |
| MKKM-MR | $51.3_{\pm 0.0}$ | $63.2_{\pm 1.5}$ | $34.7_{\pm 1.8}$ | $22.9_{\pm 0.4}$ | $50.3_{\pm 0.8}$ | $39.7_{\pm 0.5}$ | $38.4_{\pm 1.1}$ | $21.2_{\pm 0.9}$ | $40.2_{\pm 0.4}$ | $46.2_{\pm 1.4}$ |
| LKAM | $60.0_{\pm 0.0}$ | $73.9_{\pm 0.5}$ | $37.7_{\pm 1.2}$ | $27.1_{\pm 0.1}$ | $47.6_{\pm 0.0}$ | $40.5_{\pm 0.4}$ | $41.4_{\pm 0.5}$ | $20.4_{\pm 0.3}$ | $41.4_{\pm 0.8}$ | $45.5_{\pm 0.0}$ |
| LFMVC | $54.5_{\pm 0.0}$ | $76.4_{\pm 2.9}$ | $33.0_{\pm 1.4}$ | $26.4_{\pm 0.5}$ | $39.2_{\pm 0.1}$ | $59.5_{\pm 0.6}$ | $45.8_{\pm 1.0}$ | $25.1_{\pm 0.5}$ | $38.4_{\pm 1.2}$ | $45.7_{\pm 1.6}$ |
| NKSS | $55.9_{\pm 0.0}$ | $64.1_{\pm 1.2}$ | $36.4_{\pm 0.7}$ | $25.5_{\pm 0.6}$ | $39.2_{\pm 0.1}$ | $48.2_{\pm 1.0}$ | $40.4_{\pm 0.3}$ | $20.0_{\pm 0.2}$ | $41.7_{\pm 0.8}$ | $37.7_{\pm 1.4}$ |
| SPMKC | $54.5_{\pm 0.0}$ | $51.3_{\pm 1.9}$ | $17.8_{\pm 0.5}$ | $26.3_{\pm 0.2}$ | $51.4_{\pm 0.1}$ | $25.0_{\pm 0.6}$ | $38.0_{\pm 0.1}$ | $16.2_{\pm 0.2}$ | $25.6_{\pm 0.4}$ | $26.8_{\pm 0.0}$ |
| HMKC | $55.4_{\pm 0.0}$ | $91.1_{\pm 3.7}$ | $35.3_{\pm 1.5}$ | $22.4_{\pm 0.4}$ | $32.7_{\pm 0.5}$ | $64.2_{\pm 0.1}$ | $50.5_{\pm 0.1}$ | $32.8_{\pm 0.5}$ | $47.7_{\pm 1.3}$ | $46.8_{\pm 0.3}$ |
| SMKKM | $53.9_{\pm 0.0}$ | $64.2_{\pm 1.6}$ | $34.7_{\pm 1.9}$ | $22.4_{\pm 0.4}$ | $49.5_{\pm 0.5}$ | $41.5_{\pm 0.0}$ | $43.6_{\pm 1.0}$ | $22.2_{\pm 0.7}$ | $42.5_{\pm 0.8}$ | $45.5_{\pm 0.7}$ |
| OPLFMVC | $54.6_{\pm 0.1}$ | $89.2_{\pm 3.2}$ | $31.1_{\pm 2.6}$ | $27.3_{\pm 1.0}$ | $47.3_{\pm 3.1}$ | $46.1_{\pm 2.3}$ | $43.9_{\pm 1.8}$ | $23.7_{\pm 0.9}$ | $30.4_{\pm 1.0}$ | $43.9_{\pm 1.0}$ |
| LSMKKM | $58.3_{\pm 0.0}$ | $73.4_{\pm 1.6}$ | $36.3_{\pm 1.5}$ | $24.8_{\pm 0.2}$ | $57.1_{\pm 0.8}$ | $45.7_{\pm 0.1}$ | $44.5_{\pm 1.6}$ | $21.5_{\pm 0.9}$ | $43.8_{\pm 1.0}$ | $47.1_{\pm 1.0}$ |
| AIMC | $52.8_{\pm 0.0}$ | $70.4_{\pm 0.0}$ | $33.6_{\pm 0.0}$ | $25.5_{\pm 0.0}$ | $47.9_{\pm 0.0}$ | $45.4_{\pm 0.0}$ | $44.5_{\pm 0.0}$ | $24.5_{\pm 0.0}$ | $41.0_{\pm 0.0}$ | $43.2_{\pm 0.0}$ |
| OMSC | $53.0_{\pm 0.0}$ | $89.0_{\pm 0.0}$ | $31.8_{\pm 0.0}$ | $28.1_{\pm 0.0}$ | $56.5_{\pm 0.0}$ | $39.5_{\pm 0.0}$ | $41.7_{\pm 0.0}$ | $25.1_{\pm 0.0}$ | $38.9_{\pm 0.0}$ | $42.4_{\pm 0.0}$ |
| HFLSMKKM | $57.4_{\pm 0.0}$ | $51.6_{\pm 1.3}$ | $33.8_{\pm 1.1}$ | $24.2_{\pm 0.3}$ | $43.6_{\pm 0.1}$ | $31.3_{\pm 0.6}$ | $41.7_{\pm 0.4}$ | $18.5_{\pm 0.3}$ | $35.8_{\pm 0.8}$ | $37.5_{\pm 0.8}$ |
| GMC | $51.0_{\pm 0.2}$ | $88.2_{\pm 0.0}$ | $29.3_{\pm 0.0}$ | $21.2_{\pm 0.5}$ | $39.4_{\pm 0.0}$ | $25.2_{\pm 0.0}$ | $26.9_{\pm 0.6}$ | $16.8_{\pm 0.4}$ | $34.1_{\pm 0.0}$ | - |
| LTBPL | $58.3_{\pm 0.0}$ | $96.5_{\pm 0.0}$ | $32.1_{\pm 1.1}$ | $28.8_{\pm 0.0}$ | $48.2_{\pm 0.0}$ | $29.1_{\pm 0.0}$ | $40.1_{\pm 0.7}$ | - | - | - |
| UGLTL | $53.6_{\pm 0.0}$ | $99.1_{\pm 0.2}$ | $51.1_{\pm 1.7}$ | $37.1_{\pm 2.0}$ | $68.6_{\pm 1.2}$ | $92.2_{\pm 0.0}$ | $94.4_{\pm 1.3}$ | $43.7_{\pm 1.3}$ | $65.8_{\pm 2.3}$ | - |
| WTNNM | $53.3_{\pm 0.0}$ | $95.2_{\pm 0.0}$ | $43.2_{\pm 1.7}$ | $32.0_{\pm 0.2}$ | $68.0_{\pm 0.1}$ | $64.8_{\pm 0.0}$ | $76.1_{\pm 1.2}$ | $47.7_{\pm 0.0}$ | $61.7_{\pm 0.9}$ | - |
| KCGT | $54.8_{\pm 0.2}$ | $74.4_{\pm 1.2}$ | $33.4_{\pm 1.2}$ | $26.1_{\pm 0.4}$ | $52.4_{\pm 0.6}$ | $44.9_{\pm 0.4}$ | $45.5_{\pm 0.9}$ | $23.9_{\pm 0.5}$ | $39.5_{\pm 0.8}$ | $43.0_{\pm 0.8}$ |
| DLEFT-MKC | $86.4_{\pm 0.0}$ | $99.2_{\pm 0.1}$ | $66.5_{\pm 2.9}$ | $84.9_{\pm 0.4}$ | $94.1_{\pm 0.1}$ | $96.0_{\pm 0.0}$ | $96.2_{\pm 0.1}$ | $81.5_{\pm 2.7}$ | $79.9_{\pm 2.2}$ | $97.0_{\pm 4.0}$ |

Table 2: Empirical comparison of the proposed DLEFT-MKC with dozens of recent MKC algorithms on ten benchmark datasets in terms of ACC. The best result is bolded and highlighted in red, the second-best and third-best ones are represented in blue and orange, respectively.

nificant performance enhancement by surpassing UGLTL and WTNNM in terms of ACC by 32.8%, 0.1%, 15.4%, 47.8%, 25.5%, 3.8%, 1.8%, 37.8%, 14.1% as well as 33.1%, 4%, 23.3%, 52.9%, 26.1%, 31.2%, 20.1%, 33.8%, 18.2% across all datasets.

(3) The comparative analysis of various algorithms reveals that while late fusion strategies have improved clustering performance, they are not without limitations; for instance, LFMVC and OPLFMVC showed reduced complexity but struggled with unstable performance due to their heavy reliance on initial base partitions too much. Through dynamic restructuring of partitions, DLEFT-MKC significantly enhances performance; it exceeds LFMVC and OPLFMVC by 43.1% and 44.4%, respectively when averaged over ten datasets.

In summary, these results validate the effectiveness of our proposed DLEFT-MKC in enhancing clustering performance across multiple datasets; significant improvements in ACC—averaging around 10% over existing state-of-the-art algorithms—underscore its potential as a leading solution in multi-view clustering research domains. By leveraging tensor learning alongside a min-max optimization framework, our approach addresses existing challenges and sets a new benchmark for future research in this domain. DLEFT-MKC's ability to maintain high accuracy while significantly reducing computational complexity demonstrates its robustness and efficiency, qualities that are particularly essential for real-world applications requiring large-scale data processing.

### 4.2.2 EVOLUTION AND CONVERGENCE

We calculate the error value and clustering performance at each iteration to analyze the evolution of DLEFT-MKC, as illustrated in Fig. 1. As observed, the error curve initially oscillates, followed by a sharp decrease, and ultimately converges rapidly. The corresponding clustering performance improves significantly during the initial oscillation phase (learning process) before stabilizing, thereby effectively demonstrating both the necessity and efficacy of the learning process. This analysis highlights how DLEFT-MKC adapts over iterations, leading to enhanced clustering results.

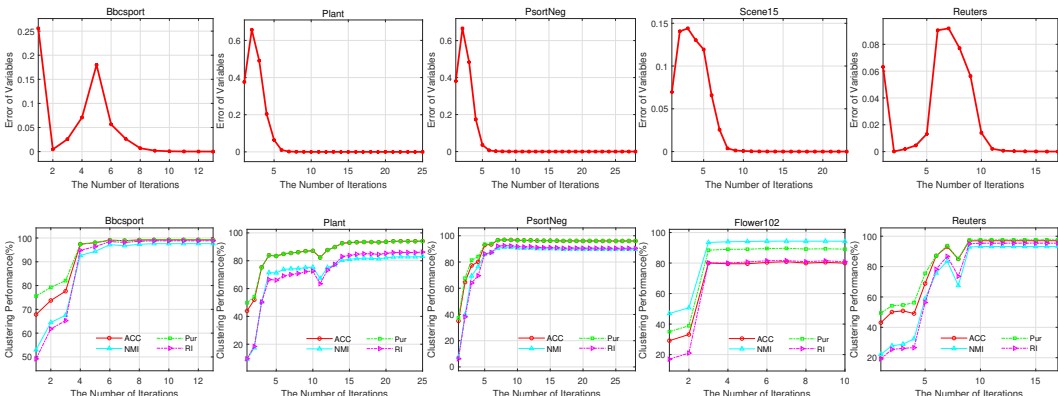

Figure 1: The evolution of error values and clustering performance during the clustering learning process of our proposed DLEFT-MKC across iterations.

### 4.2.3 CLUSTER PARTITIONS ANALYSIS

We further analyze the learned cluster partition and illustrate the visual results in Figure 2. As observed, through the learning process of DLEFT-MKC, the cluster partition becomes increasingly clear and distinguishable, manifested as a more pronounced block diagonal structure. This observation further reinforces the effectiveness of our proposed DLEFT-MKC in achieving well-defined clusters. These results demonstrate that DLEFT-MKC not only enhances clustering performance but also facilitates better interpretability of the clustered data.

### 4.2.4 PARAMETERS SENSITIVITY ANALYSIS

In order to further investigate the impact of two parameters on DLEFT-MKC, we conducted a separate experiment to analyze their sensitivity and effectiveness, as illustrated in Figure 3. As shown, two trade-off parameters introduced by DLEFT-MKC exert significant effects, demonstrating regularity and consistency across various datasets. This indicates that each term in Eq.(6) plays a crucial role, suggesting that DLEFT-MKC exhibits stability within small ranges of parameters while main-

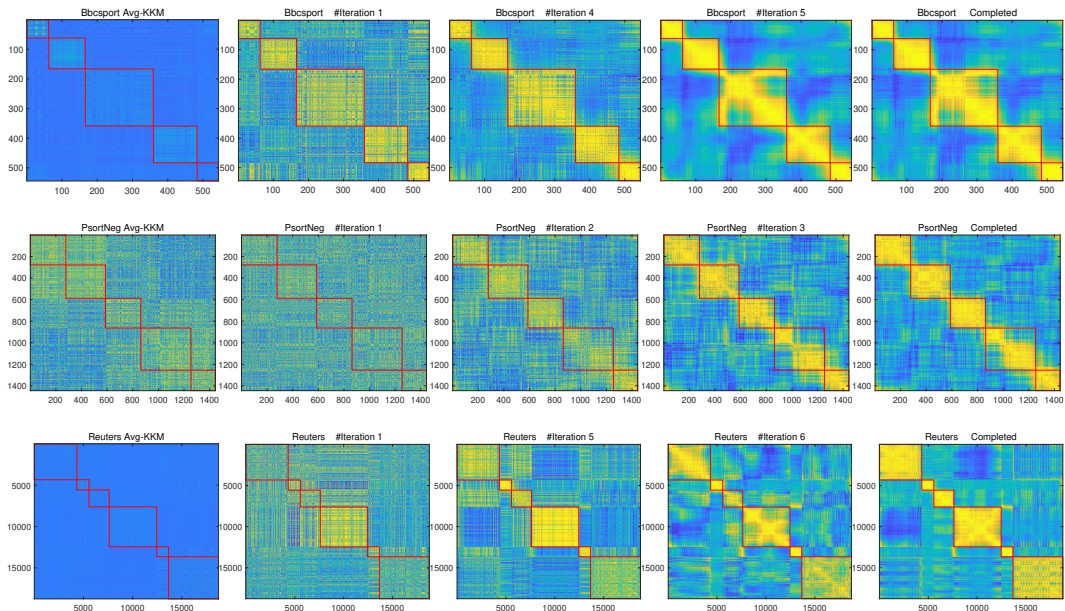

Figure 2: The leftmost figure denotes the clustering partition learned by avg-KKM. The four right figures represent the clustering partitions of DLEFT-MKC during the learning process.

taining good generalization ability. In addition, this analysis can guide the adjustment strategies for DLEFT-MKC across different datasets.

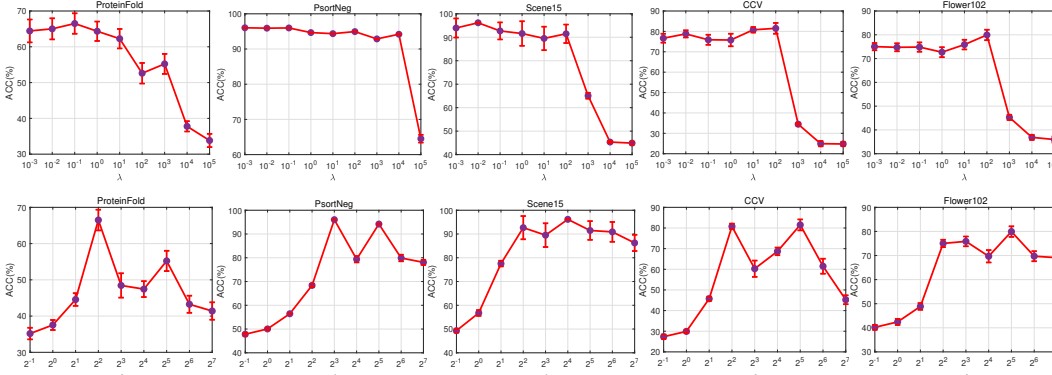

Figure 3: The effect on clustering performance with varying parameter $\lambda$ ($1^{st}$ line) and $\rho$ ($2^{nd}$ line) of the proposed DLEFT-MKC.

### 4.2.5 ABLATION STUDY

To investigate the factors contributing to the superior performance of the proposed DLEFT-MKC, we conducted a series of ablation experiments focusing on four key components: the active reconstruction of the base partitions, tensor learning guidance, the alignment strategy utilizing permutation matrices, and the min-max optimization paradigm. Specifically in Tab.3, $\mathcal{L}_1$ denotes $\rho = \lambda = 0$, $\mathcal{L}_2$ denotes $\rho = 0$, $\mathcal{L}_3$ denotes $\lambda = 0$, $\mathcal{L}_4$ denotes the exclusion of the permutation, and $\mathcal{L}_5$ denotes the absence of the min-max paradigm. As shown, our proposed DLEFT-MKC always achieves either superior or competitive performance, while $\mathcal{L}_5$, although performing well but lacking stability, indicates the robustness afforded by the min-max paradigm. Additionally, $\mathcal{L}_4$ also demonstrates a significant performance drop, highlighting the importance of the permutation matrix. Furthermore, $\mathcal{L}_1$, $\mathcal{L}_2$, and $\mathcal{L}_3$ remain inferior to DLEFT-MKC, thereby underscoring the efficacy of our proposed dynamic restriction late fusion strategy utilizing tensor learning. These findings collectively validate the effectiveness of our proposed algorithm and underscore the importance of integrating these components for optimal clustering performance.

| Algorithms | Liver | BBCSport | ProteinFold | Willow | Plant | PsortNeg | Scene15 | CCV | Flower102 | Reuters |
|---|---|---|---|---|---|---|---|---|---|---|
| $\mathcal{L}_1$ | $54.1_{\pm 0.2}$ | $63.4_{\pm 1.3}$ | $30.0_{\pm 2.1}$ | $22.2_{\pm 0.3}$ | $56.0_{\pm 0.5}$ | $38.5_{\pm 0.6}$ | $43.8_{\pm 1.6}$ | $19.6_{\pm 0.6}$ | $27.2_{\pm 0.9}$ | $45.1_{\pm 0.3}$ |
| $\mathcal{L}_2$ | $62.3_{\pm 0.0}$ | $84.8_{\pm 8.7}$ | $\mathbf{66.7}_{\pm 2.9}$ | $71.2_{\pm 0.5}$ | $93.7_{\pm 0.0}$ | $95.8_{\pm 0.0}$ | $93.8_{\pm 4.3}$ | $77.2_{\pm 2.5}$ | $74.6_{\pm 2.1}$ | $96.9_{\pm 0.0}$ |
| $\mathcal{L}_3$ | $51.0_{\pm 0.0}$ | $43.1_{\pm 0.7}$ | $13.7_{\pm 0.7}$ | $19.5_{\pm 0.6}$ | $31.9_{\pm 1.0}$ | $44.4_{\pm 3.2}$ | $54.6_{\pm 2.7}$ | $25.5_{\pm 1.1}$ | $60.6_{\pm 1.5}$ | $51.7_{\pm 0.0}$ |
| $\mathcal{L}_4$ | $82.3_{\pm 0.0}$ | $86.8_{\pm 0.3}$ | $44.6_{\pm 2.4}$ | $79.2_{\pm 2.5}$ | $72.7_{\pm 0.1}$ | $89.6_{\pm 0.0}$ | $86.3_{\pm 5.2}$ | $73.0_{\pm 1.7}$ | $80.4_{\pm 1.8}$ | $87.1_{\pm 2.2}$ |
| $\mathcal{L}_5$ | $60.6_{\pm 0.0}$ | $96.6_{\pm 0.1}$ | $56.1_{\pm 2.3}$ | $76.7_{\pm 4.3}$ | $91.0_{\pm 0.0}$ | $94.6_{\pm 0.0}$ | $95.1_{\pm 2.9}$ | $72.4_{\pm 1.6}$ | $\mathbf{81.7}_{\pm 2.7}$ | $94.8_{\pm 1.7}$ |
| **Proposed** | $\mathbf{86.4}_{\pm 0.0}$ | $\mathbf{99.2}_{\pm 0.1}$ | $66.5_{\pm 2.9}$ | $\mathbf{84.9}_{\pm 0.4}$ | $\mathbf{94.1}_{\pm 0.1}$ | $\mathbf{96.0}_{\pm 0.0}$ | $\mathbf{96.2}_{\pm 0.1}$ | $81.5_{\pm 2.7}$ | $79.9_{\pm 2.2}$ | $\mathbf{97.0}_{\pm 4.0}$ |

Table 3: Ablation study of the proposed DLEFT-MKC. The best result are highlighted in bold.

### 4.2.6 RUNNING TIME COMPARISON

Finally, to evaluate the complexity of the algorithms, we report the time consumption in Fig. 4 and Tab. 9 in the appendix. The analysis of running times across various clustering algorithms reveals significant disparities in computational efficiency. For instance, LFMVC takes 77.1s and 41.4s on Flower102 and Reuters; in contract, tensor-based clustering like WTNNM and KCGT exhibit markedly higher running times, with taking 34717s and 5976s on Flower102 and KCGT reaching 11424s on Reuters. This stark difference highlights the computational demands associated with existing tensor learning approaches, which seriously undermines their application prospects in real-world scenarios. Notably, our proposed DLEFT-MKC addresses this important problem caused by tensor learning while delivering superior clustering accuracy in less than a minute on the same datasets. Overall, DLEFT-MKC not only demonstrates advanced clustering performance but also significantly reduces computational overhead, thereby validating its effectiveness for large-scale clustering tasks.

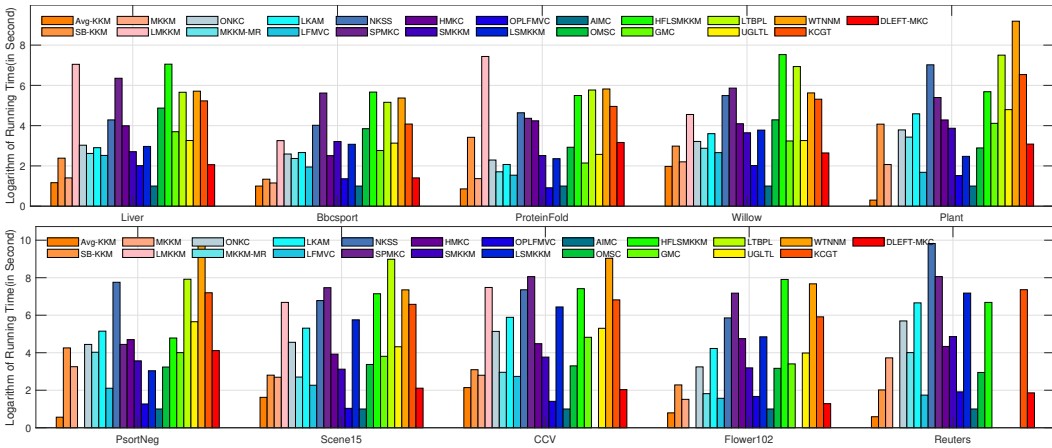

Figure 4: Time complexity comparison of all algorithms on benchmark datasets. For better clarity, we scaled the values and adopted logarithmic values in second.

### 4.2.7 ADDITIONAL EXPERIMENT

To further validate the robustness of DLEFT-MKC, we conducted additional experiments with noisy data. The visual comparisons of clustering results are presented in Fig.7 in the appendix. As shown, DLEFT-MKC not only outperforms baseline algorithms in terms of clustering performance but also shows minimal fluctuation when subjected to noise interference, highlighting its superior robustness.

## 5 CONCLUSION

This paper introduces a novel Multiple Kernel clustering framework known as **D**ynamic **L**at**E**-**F**usion Multiple Kernel Clustering with Robust **T**ensor Learning (DLEFT-MKC) via min-max optimization, which is simple yet effective and efficient. Specifically, For the first time, DLEFT-MKC integrates a min-max optimization paradigm into tensor-based MKC, enhancing both performance and robustness; the framework dynamically reconstructs base partitions from LFMVC, effectively overcoming their representational bottleneck. Additionally, tensor learning is employed to capture the high-order correlations and uncover latent structures across views. To solve the resultant optimization problem, we design an innovative and efficient strategy to combine the RGDM with ADMM. Experimental results demonstrate that our proposed DLEFT-MKC significantly outperforms other state-of-the-art MKC algorithms in terms of clustering performance and computation efficiency across benchmark datasets.

## ACKNOWLEDGEMENTS

This work is supported by National Science and Technology Innovation 2030 Major Project under Grant No. 2022ZD0209103; National Natural Science Foundation of China under Grant No. 62441618, 62421002, 62325604, 62276271, 62476281; National Natural Science Foundation of China Joint Found under Grant No. U24A20323.

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

# A    APPENDIX

We include the main notations used in this manuscript 4, Algorithm 1 (Min-Max Optimization For $\boldsymbol{\gamma}$ and $\boldsymbol{F}^*$), Algorithm 2 (DLEFT-MKC), the complete experimental results in terms of ACC (Tab. 5), NMI (Tab. 6), PUR (Tab. 7) and RI (Tab. 8), running time comparison (Tab. 9), clustering evolution (Fig. 5), parameter analysis (Fig. 6), noise experiments (Fig. 7) and theframework diagram (Fig. 8) in the appendix due to the space limitation of the paper.

| $\top Notation$ | Explanation | Notation | Explanation |
|---|---|---|---|
| $n, k, m$ | The number of samples, clusters, and views | $\boldsymbol{K}_p \in \mathbb{R}^{n \times n}$ | The $p$-th base kernel matrix |
| $\rho, \lambda$ | The trade-off parameters | $\boldsymbol{F}^* \in \mathbb{R}^{n \times k}$ | The consensus partition matrix |
| $\boldsymbol{x}_i$ | The $i$-th data sample | $\boldsymbol{F}_p \in \mathbb{R}^{n \times k}$ | The $p$-th base partition matrix |
| $\sigma_i(\cdot)$ | The $i$-th largest singular value | $\hat{\boldsymbol{F}}_p \in \mathbb{R}^{n \times k}$ | The $p$-th reconstructed partition matrix |
| $\|\cdot\|_\circledast$ | The t-SVD based tensor nuclear norm | $\boldsymbol{T}_p \in \mathbb{R}^{k \times k}$ | The $p$-th perturbation matrix |
| $\boldsymbol{\gamma} \in \mathbb{R}^m$ | The kernel weight coefficient | $\boldsymbol{I}_n \in \mathbb{R}^{n \times n}$ | The $n$-th order identity matrix |
| $\Delta$ | $\{\boldsymbol{\gamma} \in \mathbb{R}^m \mid \sum_{p=1}^m \boldsymbol{\gamma}_p = 1, \gamma_q \geq 0, \forall p\}$ | $\hat{\mathsf{F}} \in \mathbb{R}^{n \times k \times m}$ | The tensor by stacking the matrices $\hat{\boldsymbol{F}}$ |
| $\nabla$ | $\{\boldsymbol{\gamma} \in \mathbb{R}^m \mid \sum_{p=1}^m \gamma_p^2 = 1, \gamma_p \geq 0, \forall p\}$ | $\mathsf{A} \in \mathbb{R}^{n \times k \times m}$ | The auxiliary tensor variable |

Table 4: Main notations used in this manuscript.

---

**Algorithm 1** Min-Max Optimization For $\boldsymbol{\gamma}$ and $\boldsymbol{F}^*$

**Input**: $\boldsymbol{F}^*, \{\hat{\boldsymbol{F}}_p, \boldsymbol{F}_p, \boldsymbol{T}_p\}_{p=1}^m, \boldsymbol{\gamma}, k, \lambda$.
**Ouput**: Weight coefficients $\boldsymbol{\gamma}$ and consensus clustering partition $\boldsymbol{F}^*$.

1: **while** not converge **do**
2:     calculate $\mathbf{H}^{(t)}$ via a kernel k-means with $\boldsymbol{K}_{\boldsymbol{\gamma}^{(t)}}$.
3:     Calculate the reduced gradient $\left[\bigtriangledown \mathcal{G}(\boldsymbol{\gamma})\right]_p$ via Eq.(13).
4:     Calculate the descent direction $\boldsymbol{V}$ in Eq.(14).
5:     Update weight coefficients $\boldsymbol{\gamma} \leftarrow \boldsymbol{\gamma} + \alpha \boldsymbol{V}$ with the step size $\alpha$.
6:     **if** $\max |\boldsymbol{\gamma} - \boldsymbol{\gamma}_{old}| \leq 10^{-4}$ **then**
7:         Converge.
8:     **end if**
9: **end while**

---

**Algorithm 2** DLEFT-MKC

**Input**: Base partition matrices $\{\boldsymbol{F}_p\}_{p=1}^m$, the number of clusters $k$, trade-off parameters $\lambda$ and $\rho$.
**Output**: Consensus clustering partition $\boldsymbol{F}^*$.

1: Initialize $\hat{\mathsf{F}} = \Phi(\boldsymbol{F}_1, \ldots, \boldsymbol{F}_m), \hat{\boldsymbol{F}}_p = \boldsymbol{F}_p, \boldsymbol{T}_p = \mathbf{I}, \gamma_p = \frac{1}{m}, \forall p, \mathsf{A} = \hat{\mathsf{F}}, \mathsf{Y} = \mathbf{0}, \mu = 0.1, \tau = 2$.
2: Calculate $\boldsymbol{F}^* = \arg \max_{\boldsymbol{F}^{*\top} \boldsymbol{F}^* = \boldsymbol{I}} \operatorname{Tr} \left( \boldsymbol{F}^* \left( \sum_{p=1}^m \gamma_p^2 \hat{\boldsymbol{F}}_p \boldsymbol{T}_p \right) \right)$.
3: **while** not converge **do**
4:     Update reconstructed partitions $\{\hat{\boldsymbol{F}}_p\}_{p=1}^m$ via Eq.(10).
5:     Update weight coefficients $\boldsymbol{\gamma}$ and consensus parition $\boldsymbol{F}^*$ by solving Algorithm 1.
6:     Update permutation matrices $\{\boldsymbol{T}_p\}_{p=1}^m$ by solving Eq.(16).
7:     Update auxiliary tensor $\mathsf{A}$ by solving Eq.(15).
8:     Update the Lagrange multiplier $\mathsf{Y}$ and the penalty factor $\mu$ via Eq.(17).
9: **end while**

---

Table 5: Empirical comparison of the proposed DLEFT-MKC with dozens of recent MKC algorithms on ten benchmark datasets in terms of ACC. The best result is bolded and highlighted in red, the second-best and third-best ones are represented in blue and orange, respectively.

| Algorithms | Liver | BBCSport | ProteinFold | Willow | Plant | PsortNeg | Scene15 | CCV | Flower102 | Reuters |
|---|---|---|---|---|---|---|---|---|---|---|
| Avg-KKM | $54.2 \pm 0.0$ | $63.2 \pm 1.4$ | $29.0 \pm 1.5$ | $22.2 \pm 0.3$ | $61.3 \pm 0.9$ | $41.0 \pm 1.4$ | $43.2 \pm 1.8$ | $19.6 \pm 0.6$ | $27.1 \pm 0.8$ | $45.5 \pm 1.5$ |
| SB-KKM | $57.9 \pm 0.1$ | $71.4 \pm 0.1$ | $33.8 \pm 1.3$ | $26.8 \pm 0.3$ | $51.2 \pm 1.1$ | $55.3 \pm 0.0$ | $39.3 \pm 0.2$ | $20.1 \pm 0.2$ | $33.0 \pm 1.0$ | $47.2 \pm 0.0$ |
| MKKM | $55.0 \pm 0.3$ | $63.0 \pm 1.5$ | $27.0 \pm 1.1$ | $22.0 \pm 0.2$ | $56.1 \pm 0.6$ | $51.9 \pm 0.3$ | $41.2 \pm 0.1$ | $18.0 \pm 0.5$ | $22.4 \pm 0.5$ | $45.4 \pm 1.5$ |
| LMKKM | $53.7 \pm 1.1$ | $63.9 \pm 1.4$ | $22.4 \pm 0.7$ | $22.6 \pm 0.2$ | - | - | $40.9 \pm 0.1$ | $18.6 \pm 0.1$ | - | - |
| ONKC | $52.9 \pm 1.9$ | $63.4 \pm 1.4$ | $36.3 \pm 1.5$ | $22.6 \pm 0.4$ | $41.4 \pm 0.2$ | $40.2 \pm 0.6$ | $39.9 \pm 1.4$ | $22.4 \pm 0.3$ | $39.5 \pm 0.7$ | $41.8 \pm 1.2$ |
| MKKM-MR | $51.3 \pm 0.0$ | $63.2 \pm 1.5$ | $34.7 \pm 1.8$ | $22.9 \pm 0.4$ | $50.3 \pm 0.8$ | $39.7 \pm 0.5$ | $38.4 \pm 1.1$ | $21.2 \pm 0.9$ | $40.2 \pm 0.9$ | $46.2 \pm 1.4$ |
| LKAM | $60.0 \pm 0.0$ | $73.9 \pm 0.5$ | $37.7 \pm 1.2$ | $27.1 \pm 0.1$ | $47.6 \pm 0.0$ | $40.5 \pm 0.4$ | $41.4 \pm 0.5$ | $20.4 \pm 0.3$ | $41.4 \pm 0.8$ | $45.5 \pm 0.0$ |
| LFMVC | $54.5 \pm 0.0$ | $76.4 \pm 2.9$ | $33.0 \pm 1.4$ | $26.4 \pm 0.5$ | $59.5 \pm 0.6$ | $45.5 \pm 0.3$ | $45.8 \pm 1.0$ | $25.1 \pm 0.5$ | $38.4 \pm 1.2$ | $45.7 \pm 1.6$ |
| NKSS | $55.9 \pm 0.0$ | $64.1 \pm 1.2$ | $36.4 \pm 0.7$ | $25.5 \pm 0.6$ | $39.2 \pm 0.1$ | $48.2 \pm 1.0$ | $40.4 \pm 0.3$ | $20.0 \pm 0.2$ | $41.7 \pm 0.8$ | $37.7 \pm 1.4$ |
| SPMKC | $54.5 \pm 0.0$ | $51.3 \pm 1.9$ | $17.8 \pm 0.5$ | $26.3 \pm 0.2$ | $51.4 \pm 0.1$ | $25.0 \pm 0.6$ | $38.0 \pm 0.1$ | $16.2 \pm 0.2$ | $25.6 \pm 0.4$ | $26.8 \pm 0.0$ |
| HMKC | $55.4 \pm 0.0$ | $91.1 \pm 3.7$ | $35.3 \pm 1.5$ | $32.7 \pm 0.5$ | $64.2 \pm 0.1$ | $49.1 \pm 0.0$ | $50.5 \pm 0.1$ | $32.8 \pm 0.5$ | $47.7 \pm 1.3$ | $46.8 \pm 0.3$ |
| SMKKM | $53.9 \pm 0.0$ | $64.2 \pm 1.6$ | $34.7 \pm 1.9$ | $22.4 \pm 0.4$ | $49.5 \pm 0.5$ | $41.5 \pm 0.0$ | $43.6 \pm 1.0$ | $22.2 \pm 0.7$ | $42.5 \pm 0.8$ | $45.5 \pm 0.7$ |
| OPLFMVC | $54.6 \pm 0.1$ | $89.2 \pm 3.2$ | $31.1 \pm 2.6$ | $27.3 \pm 1.0$ | $47.3 \pm 3.1$ | $46.1 \pm 2.3$ | $43.9 \pm 1.8$ | $23.7 \pm 0.9$ | $30.4 \pm 1.0$ | $43.9 \pm 1.0$ |
| LSMKKM | $58.3 \pm 0.0$ | $73.4 \pm 1.0$ | $36.3 \pm 1.5$ | $24.8 \pm 0.2$ | $57.1 \pm 0.8$ | $45.7 \pm 0.1$ | $44.5 \pm 1.6$ | $21.5 \pm 0.9$ | $43.8 \pm 1.0$ | $47.1 \pm 1.0$ |
| AIMC | $52.8 \pm 0.0$ | $70.4 \pm 0.0$ | $33.6 \pm 0.0$ | $25.5 \pm 0.0$ | $47.9 \pm 0.0$ | $45.4 \pm 0.0$ | $44.5 \pm 0.0$ | $24.5 \pm 0.0$ | $41.0 \pm 0.0$ | $43.2 \pm 0.0$ |
| OMSC | $53.0 \pm 0.0$ | $89.0 \pm 0.0$ | $31.8 \pm 0.0$ | $28.1 \pm 0.0$ | $56.5 \pm 0.0$ | $39.5 \pm 0.0$ | $41.7 \pm 0.0$ | $25.1 \pm 0.0$ | $38.9 \pm 0.0$ | $42.4 \pm 0.0$ |
| HFLSMKKM | $57.4 \pm 0.0$ | $51.6 \pm 1.3$ | $33.8 \pm 1.1$ | $24.2 \pm 0.5$ | $43.6 \pm 0.1$ | $31.3 \pm 0.6$ | $41.7 \pm 0.4$ | $18.5 \pm 0.3$ | $35.8 \pm 0.8$ | $37.5 \pm 0.8$ |
| GMC | $51.0 \pm 0.2$ | $88.2 \pm 0.0$ | $29.3 \pm 0.0$ | $21.2 \pm 0.5$ | $39.4 \pm 0.0$ | $25.2 \pm 0.0$ | $26.9 \pm 0.6$ | $16.8 \pm 0.4$ | $34.1 \pm 0.0$ | - |
| LTBPL | $58.3 \pm 0.0$ | $96.5 \pm 0.0$ | $32.1 \pm 1.1$ | $28.8 \pm 0.0$ | $48.2 \pm 0.0$ | $29.1 \pm 0.0$ | $40.1 \pm 0.7$ | - | - | - |
| UGLTL | $53.6 \pm 0.0$ | $99.1 \pm 0.2$ | $51.1 \pm 1.7$ | $37.1 \pm 2.0$ | $68.6 \pm 1.2$ | $92.2 \pm 0.0$ | $94.4 \pm 5.1$ | $43.7 \pm 1.3$ | $65.8 \pm 2.3$ | - |
| WTNNM | $53.3 \pm 0.0$ | $95.2 \pm 0.0$ | $43.2 \pm 1.7$ | $32.0 \pm 0.2$ | $68.0 \pm 0.1$ | $64.8 \pm 0.0$ | $76.1 \pm 1.2$ | $47.7 \pm 0.0$ | $61.7 \pm 0.9$ | - |
| KCGT | $54.8 \pm 0.2$ | $74.4 \pm 1.2$ | $33.4 \pm 1.2$ | $26.1 \pm 0.4$ | $52.4 \pm 0.6$ | $44.9 \pm 0.4$ | $45.5 \pm 0.9$ | $23.9 \pm 0.5$ | $39.5 \pm 0.8$ | $43.0 \pm 0.8$ |
| DLEFT-MKC | $\mathbf{86.4 \pm 0.0}$ | $\mathbf{99.2 \pm 0.1}$ | $\mathbf{66.5 \pm 2.9}$ | $\mathbf{84.9 \pm 0.4}$ | $\mathbf{94.1 \pm 0.1}$ | $\mathbf{96.0 \pm 0.0}$ | $\mathbf{96.2 \pm 0.1}$ | $\mathbf{81.5 \pm 2.7}$ | $\mathbf{79.9 \pm 2.2}$ | $\mathbf{97.0 \pm 4.0}$ |

Table 6: Empirical comparison of the proposed DLEFT-MKC with dozens of recent MKC algorithms on ten benchmark datasets in terms of NMI. The best result is bolded and highlighted in red, the second-best and third-best ones are represented in blue and orange, respectively.

| Algorithms | Liver | BBCSport | ProteinFold | Willow | Plant | PsortNeg | Scene15 | CCV | Flower102 | Reuters |
|---|---|---|---|---|---|---|---|---|---|---|
| Avg-KKM | $1.1 \pm 0.0$ | $43.5 \pm 1.1$ | $40.3 \pm 1.3$ | $5.7 \pm 0.2$ | $26.5 \pm 0.9$ | $17.4 \pm 0.7$ | $41.3 \pm 0.7$ | $16.8 \pm 0.4$ | $46.0 \pm 0.5$ | $27.4 \pm 0.4$ |
| SB-KKM | $2.1 \pm 0.2$ | $63.2 \pm 0.2$ | $41.1 \pm 1.1$ | $7.8 \pm 0.3$ | $16.9 \pm 1.1$ | $39.1 \pm 0.0$ | $37.9 \pm 0.1$ | $17.7 \pm 0.1$ | $48.7 \pm 0.4$ | $25.5 \pm 0.0$ |
| MKKM | $0.8 \pm 0.2$ | $43.6 \pm 1.2$ | $38.0 \pm 0.6$ | $5.7 \pm 0.1$ | $19.5 \pm 0.5$ | $32.2 \pm 0.2$ | $38.6 \pm 0.1$ | $15.0 \pm 0.4$ | $42.7 \pm 0.2$ | $27.3 \pm 0.4$ |
| LMKKM | $0.7 \pm 0.3$ | $44.0 \pm 0.8$ | $34.7 \pm 0.6$ | $5.7 \pm 0.1$ | - | - | $38.8 \pm 0.1$ | $14.4 \pm 0.1$ | - | - |
| ONKC | $0.6 \pm 0.3$ | $43.5 \pm 1.1$ | $44.4 \pm 0.9$ | $6.1 \pm 0.4$ | $10.5 \pm 0.2$ | $21.0 \pm 0.7$ | $37.7 \pm 0.6$ | $18.5 \pm 0.2$ | $56.1 \pm 0.4$ | $22.3 \pm 0.4$ |
| MKKM-MR | $0.3 \pm 0.0$ | $43.5 \pm 1.1$ | $43.7 \pm 1.2$ | $6.3 \pm 0.2$ | $20.4 \pm 0.4$ | $21.6 \pm 0.4$ | $37.3 \pm 0.6$ | $18.0 \pm 0.4$ | $56.7 \pm 0.5$ | $25.3 \pm 0.7$ |
| LKAM | $2.2 \pm 0.0$ | $65.4 \pm 1.0$ | $46.2 \pm 0.6$ | $8.3 \pm 0.2$ | $13.9 \pm 0.0$ | $21.8 \pm 0.8$ | $42.1 \pm 0.1$ | $17.6 \pm 0.2$ | $56.9 \pm 0.3$ | $29.9 \pm 0.0$ |
| LFMVC | $1.2 \pm 0.0$ | $58.9 \pm 3.0$ | $41.7 \pm 1.1$ | $7.9 \pm 0.3$ | $23.4 \pm 0.8$ | $18.8 \pm 0.3$ | $42.7 \pm 0.2$ | $20.1 \pm 0.3$ | $54.9 \pm 0.4$ | $27.4 \pm 0.4$ |
| NKSS | $0.9 \pm 0.0$ | $51.3 \pm 0.4$ | $46.5 \pm 0.5$ | $5.4 \pm 0.1$ | $12.6 \pm 1.1$ | $25.9 \pm 1.3$ | $39.4 \pm 0.2$ | $16.9 \pm 0.2$ | $58.6 \pm 0.2$ | $16.8 \pm 0.8$ |
| SPMKC | $2.5 \pm 0.0$ | $29.9 \pm 3.1$ | $27.3 \pm 0.5$ | $7.1 \pm 0.1$ | $24.2 \pm 0.0$ | $3.2 \pm 0.3$ | $39.3 \pm 0.1$ | $12.1 \pm 0.1$ | $42.3 \pm 0.2$ | $0.6 \pm 0.1$ |
| HMKC | $1.0 \pm 0.0$ | $78.2 \pm 4.4$ | $45.3 \pm 1.1$ | $11.9 \pm 0.5$ | $32.9 \pm 0.4$ | $24.9 \pm 0.0$ | $45.9 \pm 0.1$ | $27.6 \pm 0.2$ | $61.5 \pm 0.4$ | $30.5 \pm 0.5$ |
| SMKKM | $0.7 \pm 0.0$ | $44.4 \pm 1.0$ | $44.4 \pm 1.1$ | $5.9 \pm 0.4$ | $16.9 \pm 0.9$ | $19.1 \pm 0.1$ | $40.6 \pm 0.6$ | $18.2 \pm 0.3$ | $58.6 \pm 0.5$ | $27.7 \pm 0.2$ |
| OPLFMVC | $0.9 \pm 0.1$ | $78.7 \pm 3.0$ | $40.0 \pm 2.0$ | $8.5 \pm 0.5$ | $13.3 \pm 1.0$ | $21.3 \pm 1.9$ | $41.3 \pm 0.8$ | $18.1 \pm 0.7$ | $47.2 \pm 0.4$ | $24.8 \pm 1.5$ |
| LSMKKM | $1.5 \pm 0.0$ | $65.0 \pm 1.4$ | $45.2 \pm 1.2$ | $6.3 \pm 0.1$ | $20.8 \pm 1.0$ | $17.0 \pm 0.0$ | $41.4 \pm 0.8$ | $17.8 \pm 0.4$ | $60.0 \pm 0.5$ | $27.0 \pm 0.6$ |
| AIMC | $0.4 \pm 0.0$ | $69.6 \pm 0.0$ | $42.9 \pm 0.0$ | $6.6 \pm 0.0$ | $13.9 \pm 0.0$ | $17.9 \pm 0.0$ | $41.6 \pm 0.0$ | $19.0 \pm 0.0$ | $54.6 \pm 0.0$ | $24.3 \pm 0.0$ |
| OMSC | $0.9 \pm 0.0$ | $73.5 \pm 0.0$ | $38.0 \pm 0.0$ | $7.7 \pm 0.0$ | $20.9 \pm 0.0$ | $12.9 \pm 0.0$ | $39.0 \pm 0.0$ | $19.1 \pm 0.0$ | $52.8 \pm 0.0$ | $24.7 \pm 0.0$ |
| HFLSMKKM | $1.9 \pm 0.0$ | $35.4 \pm 1.2$ | $44.7 \pm 0.6$ | $5.0 \pm 0.2$ | $18.6 \pm 0.1$ | $8.3 \pm 0.3$ | $44.0 \pm 0.3$ | $15.1 \pm 0.2$ | $55.1 \pm 0.3$ | $18.6 \pm 0.8$ |
| GMC | $0.1 \pm 0.0$ | $78.3 \pm 0.0$ | $25.9 \pm 0.0$ | $2.7 \pm 0.3$ | $0.8 \pm 0.0$ | $1.6 \pm 0.0$ | $18.9 \pm 0.5$ | $15.6 \pm 0.2$ | $41.9 \pm 0.0$ | - |
| LTBPL | $0.4 \pm 0.0$ | $88.6 \pm 0.0$ | $43.4 \pm 0.7$ | $7.2 \pm 0.0$ | $10.7 \pm 0.0$ | $6.0 \pm 0.0$ | $36.1 \pm 0.3$ | - | - | - |
| UGLTL | $0.3 \pm 0.0$ | $96.7 \pm 0.6$ | $73.2 \pm 1.2$ | $29.7 \pm 1.4$ | $43.3 \pm 0.7$ | $83.4 \pm 0.1$ | $94.4 \pm 2.1$ | $58.9 \pm 0.5$ | $89.4 \pm 0.5$ | - |
| WTNNM | $0.3 \pm 0.0$ | $85.1 \pm 0.0$ | $51.0 \pm 0.8$ | $13.3 \pm 0.2$ | $35.1 \pm 0.1$ | $42.6 \pm 0.0$ | $74.7 \pm 0.5$ | $38.9 \pm 0.1$ | $74.7 \pm 0.4$ | - |
| KCGT | $1.0 \pm 0.1$ | $61.0 \pm 1.2$ | $42.8 \pm 0.8$ | $8.1 \pm 0.3$ | $19.8 \pm 0.5$ | $22.8 \pm 0.4$ | $43.5 \pm 0.4$ | $20.8 \pm 0.3$ | $55.7 \pm 0.3$ | $23.8 \pm 0.4$ |
| DLEFT-MKC | $45.2 \pm 0.0$ | $97.6 \pm 0.2$ | $80.2 \pm 1.1$ | $76.5 \pm 0.4$ | $83.1 \pm 0.1$ | $89.6 \pm 0.1$ | $94.0 \pm 0.1$ | $77.4 \pm 0.7$ | $94.3 \pm 0.6$ | $93.2 \pm 2.1$ |

Table 7: Empirical comparison of the proposed DLEFT-MKC with dozens of recent MKC algorithms on ten benchmark datasets in terms of PUR. The best result is bolded and highlighted in red, the second-best and third-best ones are represented in blue and orange, respectively.

| Algorithms | Liver | BBCSport | ProteinFold | Willow | Plant | PsortNeg | Scene15 | CCV | Flower102 | Reuters |
|---|---|---|---|---|---|---|---|---|---|---|
| Avg-KKM | $58.0 \pm 0.0$ | $68.1 \pm 0.7$ | $37.4 \pm 1.7$ | $27.1 \pm 0.2$ | $61.3 \pm 0.9$ | $43.3 \pm 1.0$ | $47.8 \pm 1.4$ | $23.7 \pm 0.5$ | $32.3 \pm 0.6$ | $53.0 \pm 0.4$ |
| SB-KKM | $58.0 \pm 0.0$ | $78.8 \pm 0.1$ | $39.4 \pm 1.2$ | $30.1 \pm 0.4$ | $53.2 \pm 0.5$ | $61.6 \pm 0.0$ | $42.8 \pm 0.1$ | $23.3 \pm 0.2$ | $38.4 \pm 0.7$ | $53.9 \pm 0.0$ |
| MKKM | $58.0 \pm 0.0$ | $68.2 \pm 0.8$ | $33.7 \pm 1.1$ | $27.2 \pm 0.2$ | $56.1 \pm 0.6$ | $56.6 \pm 0.2$ | $44.3 \pm 0.2$ | $22.2 \pm 0.5$ | $27.8 \pm 0.4$ | $52.9 \pm 0.5$ |
| LMKKM | $58.0 \pm 0.0$ | $68.4 \pm 0.7$ | $31.2 \pm 1.0$ | $27.4 \pm 0.2$ | - | - | $44.3 \pm 0.2$ | $22.0 \pm 0.1$ | - | - |
| ONKC | $58.0 \pm 0.0$ | $68.1 \pm 0.7$ | $42.7 \pm 1.3$ | $27.3 \pm 0.4$ | $49.0 \pm 0.1$ | $44.7 \pm 0.3$ | $43.6 \pm 0.9$ | $24.6 \pm 0.3$ | $45.6 \pm 0.7$ | $52.6 \pm 0.3$ |
| MKKM-MR | $58.0 \pm 0.0$ | $68.0 \pm 0.7$ | $41.9 \pm 1.4$ | $27.5 \pm 0.3$ | $56.7 \pm 0.1$ | $44.7 \pm 0.4$ | $42.4 \pm 1.0$ | $23.7 \pm 0.7$ | $46.3 \pm 0.8$ | $52.2 \pm 0.6$ |
| LKAM | $60.0 \pm 0.0$ | $79.4 \pm 0.5$ | $43.7 \pm 0.8$ | $29.4 \pm 0.4$ | $54.5 \pm 0.0$ | $45.3 \pm 0.5$ | $46.0 \pm 0.3$ | $23.3 \pm 0.2$ | $48.0 \pm 0.6$ | $55.4 \pm 0.0$ |
| LFMVC | $58.0 \pm 0.0$ | $76.7 \pm 2.7$ | $39.3 \pm 1.5$ | $29.9 \pm 0.4$ | $59.5 \pm 0.6$ | $48.2 \pm 0.3$ | $49.4 \pm 0.5$ | $28.2 \pm 0.4$ | $44.6 \pm 0.8$ | $53.2 \pm 0.4$ |
| NKSS | $58.0 \pm 0.0$ | $72.6 \pm 0.2$ | $44.8 \pm 0.6$ | $27.4 \pm 0.1$ | $54.0 \pm 0.7$ | $54.3 \pm 1.6$ | $44.1 \pm 0.2$ | $23.6 \pm 0.3$ | $48.7 \pm 0.4$ | $46.9 \pm 0.2$ |
| SPMKC | $58.0 \pm 0.0$ | $56.1 \pm 1.5$ | $23.7 \pm 0.7$ | $30.8 \pm 0.2$ | $59.0 \pm 0.1$ | $27.8 \pm 0.1$ | $42.6 \pm 0.1$ | $20.8 \pm 0.3$ | $31.2 \pm 0.4$ | $27.4 \pm 0.0$ |
| HMKC | $58.0 \pm 0.0$ | $91.1 \pm 3.7$ | $42.9 \pm 1.9$ | $34.2 \pm 0.4$ | $64.2 \pm 0.1$ | $53.0 \pm 0.0$ | $53.1 \pm 0.1$ | $36.5 \pm 0.4$ | $54.5 \pm 0.8$ | $53.9 \pm 0.1$ |
| SMKKM | $58.0 \pm 0.0$ | $68.7 \pm 0.9$ | $41.8 \pm 1.5$ | $27.3 \pm 0.3$ | $54.3 \pm 0.3$ | $42.2 \pm 0.1$ | $48.4 \pm 1.3$ | $25.3 \pm 0.5$ | $48.6 \pm 0.7$ | $53.3 \pm 0.0$ |
| OPLFMVC | $58.0 \pm 0.0$ | $89.6 \pm 2.1$ | $36.4 \pm 2.6$ | $30.3 \pm 0.9$ | $50.5 \pm 2.4$ | $49.5 \pm 2.5$ | $46.8 \pm 1.7$ | $26.9 \pm 0.8$ | $34.7 \pm 0.7$ | $51.7 \pm 1.2$ |
| LSMKKM | $58.3 \pm 0.0$ | $79.2 \pm 0.5$ | $42.6 \pm 1.5$ | $27.7 \pm 0.3$ | $58.5 \pm 1.1$ | $47.2 \pm 0.1$ | $49.3 \pm 1.6$ | $24.7 \pm 0.6$ | $50.2 \pm 0.9$ | $52.9 \pm 0.2$ |
| AIMC | $58.0 \pm 0.0$ | $80.5 \pm 0.0$ | $38.9 \pm 0.0$ | $28.2 \pm 0.0$ | $55.5 \pm 0.0$ | $47.6 \pm 0.0$ | $48.6 \pm 0.0$ | $28.6 \pm 0.0$ | $44.6 \pm 0.0$ | $52.8 \pm 0.0$ |
| OMSC | $58.0 \pm 0.0$ | $89.0 \pm 0.0$ | $37.2 \pm 0.0$ | $30.4 \pm 0.0$ | $57.6 \pm 0.0$ | $43.5 \pm 0.0$ | $44.8 \pm 0.0$ | $27.9 \pm 0.0$ | $42.3 \pm 0.0$ | $49.8 \pm 0.0$ |
| HFLSMKKM | $58.0 \pm 0.0$ | $61.4 \pm 0.1$ | $41.8 \pm 0.9$ | $27.6 \pm 0.2$ | $54.7 \pm 0.0$ | $32.8 \pm 0.6$ | $46.0 \pm 0.4$ | $21.5 \pm 0.3$ | $43.7 \pm 0.6$ | $46.9 \pm 0.8$ |
| GMC | $58.0 \pm 0.0$ | $88.2 \pm 0.0$ | $32.6 \pm 0.0$ | $23.3 \pm 0.5$ | $39.7 \pm 0.0$ | $27.1 \pm 0.0$ | $27.9 \pm 0.6$ | $20.9 \pm 0.4$ | $38.9 \pm 0.0$ | - |
| LTBPL | $58.3 \pm 0.0$ | $96.5 \pm 0.0$ | $38.7 \pm 0.8$ | $30.1 \pm 0.0$ | $49.7 \pm 0.0$ | $31.2 \pm 0.1$ | $42.4 \pm 0.5$ | - | - | - |
| UGLTL | $58.0 \pm 0.0$ | $99.1 \pm 0.2$ | $64.8 \pm 2.0$ | $40.9 \pm 1.2$ | $70.2 \pm 0.3$ | $92.2 \pm 0.0$ | $95.4 \pm 3.5$ | $53.1 \pm 0.9$ | $80.4 \pm 1.1$ | - |
| WTNNM | $58.0 \pm 0.0$ | $95.2 \pm 0.0$ | $49.3 \pm 1.1$ | $36.8 \pm 0.2$ | $68.0 \pm 0.1$ | $68.1 \pm 0.1$ | $80.9 \pm 0.5$ | $49.7 \pm 0.0$ | $69.0 \pm 0.6$ | - |
| KCGT | $58.1 \pm 0.0$ | $78.2 \pm 0.8$ | $40.2 \pm 1.2$ | $29.6 \pm 0.3$ | $56.3 \pm 0.4$ | $48.0 \pm 0.4$ | $49.1 \pm 0.8$ | $27.5 \pm 0.4$ | $45.8 \pm 0.6$ | $50.5 \pm 0.3$ |
| DLEFT-MKC | $\mathbf{86.4 \pm 0.0}$ | $\mathbf{99.2 \pm 0.1}$ | $\mathbf{78.3 \pm 1.8}$ | $\mathbf{84.9 \pm 0.4}$ | $\mathbf{94.1 \pm 0.1}$ | $\mathbf{96.0 \pm 0.0}$ | $\mathbf{96.2 \pm 0.1}$ | $\mathbf{82.7 \pm 1.6}$ | $\mathbf{89.1 \pm 1.2}$ | $\mathbf{97.6 \pm 1.3}$ |

Table 8: Empirical comparison of the proposed DLEFT-MKC with dozens of recent MKC algorithms on ten benchmark datasets in terms of RI. The best result is bolded and highlighted in red, the second-best and third-best ones are represented in blue and orange, respectively.

| Algorithms | Liver | BBCSport | ProteinFold | Willow | Plant | PsortNeg | Scene15 | CCV | Flower102 | Reuters |
|---|---|---|---|---|---|---|---|---|---|---|
| Avg-KKM | $0.3 \pm 0.0$ | $39.3 \pm 1.9$ | $14.4 \pm 1.8$ | $3.1 \pm 0.1$ | $24.6 \pm 1.2$ | $13.1 \pm 0.6$ | $26.0 \pm 1.1$ | $6.6 \pm 0.2$ | $15.5 \pm 0.5$ | $21.8 \pm 1.4$ |
| SB-KKM | $2.2 \pm 0.1$ | $60.4 \pm 0.2$ | $15.1 \pm 1.2$ | $4.7 \pm 0.2$ | $13.9 \pm 0.9$ | $31.6 \pm 0.0$ | $21.4 \pm 0.1$ | $6.7 \pm 0.1$ | $18.9 \pm 0.6$ | $23.6 \pm 0.0$ |
| MKKM | $0.7 \pm 0.1$ | $39.2 \pm 2.0$ | $12.1 \pm 0.7$ | $3.2 \pm 0.1$ | $17.4 \pm 0.6$ | $26.8 \pm 0.2$ | $22.6 \pm 0.1$ | $5.7 \pm 0.2$ | $12.1 \pm 0.4$ | $21.8 \pm 1.4$ |
| LMKKM | $0.2 \pm 0.4$ | $40.3 \pm 1.5$ | $7.8 \pm 0.4$ | $3.3 \pm 0.1$ | - | - | $22.9 \pm 0.1$ | $5.6 \pm 0.0$ | - | - |
| ONKC | $0.1 \pm 0.5$ | $39.5 \pm 1.9$ | $18.0 \pm 1.1$ | $3.3 \pm 0.2$ | $9.8 \pm 0.1$ | $16.9 \pm 0.3$ | $23.5 \pm 0.9$ | $7.7 \pm 0.1$ | $24.9 \pm 0.5$ | $20.3 \pm 0.3$ |
| MKKM-MR | $-0.3 \pm 0.0$ | $39.3 \pm 1.9$ | $17.2 \pm 1.5$ | $3.4 \pm 0.2$ | $19.0 \pm 0.2$ | $16.9 \pm 0.3$ | $22.7 \pm 0.9$ | $7.2 \pm 0.3$ | $25.5 \pm 0.6$ | $23.1 \pm 0.6$ |
| LKAM | $3.6 \pm 0.0$ | $62.3 \pm 1.2$ | $20.1 \pm 1.1$ | $4.6 \pm 0.1$ | $9.1 \pm 0.0$ | $16.0 \pm 0.3$ | $24.8 \pm 0.4$ | $6.9 \pm 0.1$ | $27.2 \pm 0.6$ | $24.1 \pm 0.0$ |
| LFMVC | $0.4 \pm 0.0$ | $57.0 \pm 3.8$ | $16.1 \pm 1.5$ | $4.6 \pm 0.2$ | $21.7 \pm 0.8$ | $16.1 \pm 0.2$ | $27.3 \pm 0.4$ | $9.4 \pm 0.2$ | $25.5 \pm 1.0$ | $22.1 \pm 1.6$ |
| NKSS | $1.1 \pm 0.0$ | $44.3 \pm 0.6$ | $18.5 \pm 0.6$ | $4.3 \pm 0.1$ | $8.5 \pm 0.5$ | $19.9 \pm 0.5$ | $22.8 \pm 0.1$ | $6.2 \pm 0.2$ | $27.6 \pm 0.5$ | $13.6 \pm 0.8$ |
| SPMKC | $0.1 \pm 0.0$ | $21.8 \pm 3.5$ | $4.4 \pm 0.3$ | $4.6 \pm 0.1$ | $19.1 \pm 0.0$ | $0.1 \pm 0.2$ | $21.2 \pm 0.1$ | $4.2 \pm 0.1$ | $14.5 \pm 0.4$ | $0.1 \pm 0.1$ |
| HMKC | $0.9 \pm 0.0$ | $79.3 \pm 4.8$ | $19.0 \pm 1.6$ | $8.2 \pm 0.3$ | $31.0 \pm 0.2$ | $21.8 \pm 0.1$ | $32.5 \pm 0.1$ | $14.0 \pm 0.2$ | $34.2 \pm 1.1$ | $22.6 \pm 0.5$ |
| SMKKM | $0.3 \pm 0.0$ | $40.8 \pm 1.9$ | $17.6 \pm 1.9$ | $3.2 \pm 0.2$ | $16.9 \pm 0.8$ | $13.1 \pm 0.0$ | $25.4 \pm 0.9$ | $7.5 \pm 0.2$ | $28.5 \pm 0.8$ | $22.1 \pm 0.8$ |
| OPLFMVC | $0.5 \pm 0.1$ | $81.1 \pm 4.2$ | $15.4 \pm 2.3$ | $5.0 \pm 0.4$ | $11.3 \pm 1.8$ | $17.6 \pm 1.9$ | $26.6 \pm 1.0$ | $7.9 \pm 0.6$ | $19.4 \pm 1.0$ | $20.6 \pm 0.5$ |
| LSMKKM | $2.4 \pm 0.0$ | $61.6 \pm 1.8$ | $19.9 \pm 1.2$ | $3.3 \pm 0.1$ | $19.7 \pm 1.4$ | $13.8 \pm 0.0$ | $26.3 \pm 1.4$ | $7.3 \pm 0.3$ | $29.7 \pm 0.9$ | $21.6 \pm 0.2$ |
| AIMC | $-0.0 \pm 0.0$ | $66.1 \pm 0.0$ | $19.0 \pm 0.0$ | $3.8 \pm 0.0$ | $13.5 \pm 0.0$ | $15.1 \pm 0.0$ | $28.0 \pm 0.0$ | $9.0 \pm 0.0$ | $29.8 \pm 0.0$ | $20.0 \pm 0.0$ |
| OMSC | $-0.1 \pm 0.0$ | $74.7 \pm 0.0$ | $15.9 \pm 0.0$ | $4.9 \pm 0.0$ | $20.8 \pm 0.0$ | $9.2 \pm 0.0$ | $25.9 \pm 0.0$ | $7.8 \pm 0.0$ | $26.9 \pm 0.0$ | $17.8 \pm 0.0$ |
| HFLSMKKM | $1.9 \pm 0.0$ | $22.1 \pm 1.0$ | $18.6 \pm 1.0$ | $3.0 \pm 0.1$ | $12.0 \pm 0.0$ | $4.0 \pm 0.2$ | $25.7 \pm 0.1$ | $6.3 \pm 0.1$ | $23.2 \pm 0.6$ | $13.6 \pm 0.7$ |
| GMC | $-0.2 \pm 0.0$ | $82.6 \pm 0.0$ | $2.9 \pm 0.0$ | $0.5 \pm 0.1$ | $-0.0 \pm 0.0$ | $-0.5 \pm 0.0$ | $2.5 \pm 0.0$ | $5.6 \pm 0.2$ | $2.0 \pm 0.0$ | - |
| LTBPL | $0.2 \pm 0.0$ | $90.7 \pm 0.0$ | $15.7 \pm 0.8$ | $4.7 \pm 0.0$ | $7.5 \pm 0.0$ | $2.7 \pm 0.0$ | $23.0 \pm 0.5$ | - | - | - |
| UGLTL | $0.2 \pm 0.0$ | $97.9 \pm 0.3$ | $43.3 \pm 2.3$ | $18.3 \pm 0.9$ | $42.8 \pm 0.4$ | $82.8 \pm 0.1$ | $92.3 \pm 4.8$ | $35.8 \pm 1.1$ | $66.4 \pm 1.9$ | - |
| WTNNM | $0.2 \pm 0.0$ | $88.8 \pm 0.0$ | $24.4 \pm 1.1$ | $9.2 \pm 0.2$ | $31.6 \pm 0.1$ | $40.1 \pm 0.1$ | $67.1 \pm 1.2$ | $26.1 \pm 0.0$ | $51.2 \pm 1.1$ | - |
| KCGT | $0.7 \pm 0.1$ | $58.5 \pm 1.6$ | $16.9 \pm 1.1$ | $4.9 \pm 0.2$ | $17.5 \pm 0.5$ | $18.8 \pm 0.3$ | $29.1 \pm 0.7$ | $9.7 \pm 0.2$ | $26.5 \pm 0.7$ | $19.3 \pm 0.6$ |
| DLEFT-MKC | $52.8 \pm 0.0$ | $98.8 \pm 0.1$ | $58.4 \pm 2.7$ | $71.1 \pm 0.6$ | $86.2 \pm 0.1$ | $90.5 \pm 0.1$ | $92.7 \pm 0.1$ | $71.8 \pm 2.3$ | $80.8 \pm 2.6$ | $95.4 \pm 3.0$ |

Table 9: Running time comparison of the proposed DLEFT-MKC with dozens of recent MKC algorithms on ten benchmark datasets. As seen, our proposed DLEFT-MKC always achieves the leading computation efficiency, while other tensor-based MKC algorithms, in contrast, requires an unacceptable amount of time consumption.

| Algorithms | Liver | BBCSport | ProteinFold | Willow | Plant | PsortNeg | Scene15 | CCV | Flower102 | Reuters |
|---|---|---|---|---|---|---|---|---|---|---|
| Avg-KKM | 0.0358 | 0.1153 | 0.5382 | 0.3319 | 0.3875 | 0.7345 | 4.2126 | 13.428 | 35.725 | 13.057 |
| SB-KKM | 0.1209 | 0.1621 | 6.9803 | 0.9097 | 16.860 | 29.501 | 13.745 | 34.872 | 158.49 | 54.484 |
| MKKM | 0.0454 | 0.1344 | 0.8964 | 0.4163 | 2.2695 | 10.838 | 12.313 | 25.943 | 73.470 | 301.36 |
| LMKKM | 12.791 | 1.1007 | 387.04 | 4.3772 | - | - | 665.57 | 2797.8 | - | - |
| ONKC | 0.2302 | 0.5664 | 2.2594 | 1.1470 | 12.656 | 35.544 | 79.285 | 268.68 | 414.45 | 2159.1 |
| MKKM-MR | 0.1530 | 0.4537 | 1.2601 | 0.8178 | 8.8523 | 23.433 | 12.420 | 30.355 | 99.434 | 398.62 |
| LKAM | 0.2035 | 0.6089 | 1.8104 | 1.6871 | 28.169 | 72.125 | 168.39 | 569.29 | 1106.2 | 5656.5 |
| LFMVC | 0.1386 | 0.2971 | 1.0644 | 0.6584 | 1.5419 | 3.4399 | 8.0331 | 24.395 | 77.743 | 41.380 |
| NKSS | 0.8090 | 2.3590 | 23.620 | 11.178 | 320.96 | 977.77 | 732.98 | 2473.3 | 5642.9 | 134155 |
| SPMKC | 6.3990 | 11.701 | 17.952 | 16.217 | 63.280 | 35.600 | 1457.5 | 4984.5 | 21134 | 22978 |
| HMKC | 0.6043 | 0.5222 | 15.854 | 2.7640 | 20.778 | 45.927 | 42.189 | 139.55 | 1871.2 | 554.61 |
| SMKKM | 0.1669 | 1.0544 | 2.8167 | 1.7630 | 13.746 | 14.805 | 18.915 | 68.214 | 393.37 | 937.09 |
| OPLFMVC | 0.0834 | 0.1658 | 0.5692 | 0.3462 | 1.3099 | 1.4863 | 2.3429 | 6.4595 | 85.706 | 49.162 |
| LSMKKM | 0.2166 | 0.9182 | 2.4147 | 2.0128 | 3.4047 | 8.7318 | 262.69 | 989.30 | 2059.6 | 9531.5 |
| AIMC | 0.0303 | 0.1153 | 0.6204 | 0.1251 | 0.7790 | 1.1370 | 2.2632 | 4.2910 | 43.917 | 19.736 |
| OMSC | 1.4543 | 1.9912 | 4.2627 | 3.3521 | 5.1713 | 10.648 | 24.322 | 42.777 | 383.98 | 138.50 |
| HFLSMKKM | 12.877 | 12.267 | 55.654 | 85.690 | 84.367 | 50.044 | 1054.8 | 2621.3 | 43898 | 5800.1 |
| GMC | 0.4501 | 0.6732 | 1.9474 | 1.1743 | 17.564 | 22.933 | 37.545 | 196.31 | 486.11 | - |
| LTBPL | 3.1920 | 7.3862 | 73.129 | 47.276 | 518.88 | 1149.6 | 6638.9 | - | - | - |
| UGLTL | 0.2907 | 0.9727 | 2.9899 | 1.1987 | 34.669 | 119.22 | 62.496 | 317.14 | 867.96 | - |
| WTNNM | 3.3709 | 9.1383 | 76.943 | 12.762 | 2800.6 | 8567.7 | 1298.4 | 13255 | 34717 | - |
| KCGT | 2.0792 | 2.5097 | 32.411 | 9.3431 | 197.81 | 559.06 | 599.97 | 1443.1 | 5976.3 | 11424 |
| DLEFT-MKC | 0.0875 | 0.1730 | 5.4057 | 0.6476 | 6.2585 | 25.598 | 6.8630 | 12.042 | 58.356 | 46.609 |

Figure 5: The evolution of error values and clustering performance during the clustering learning process of our proposed DLEFT-MKC across iterations.

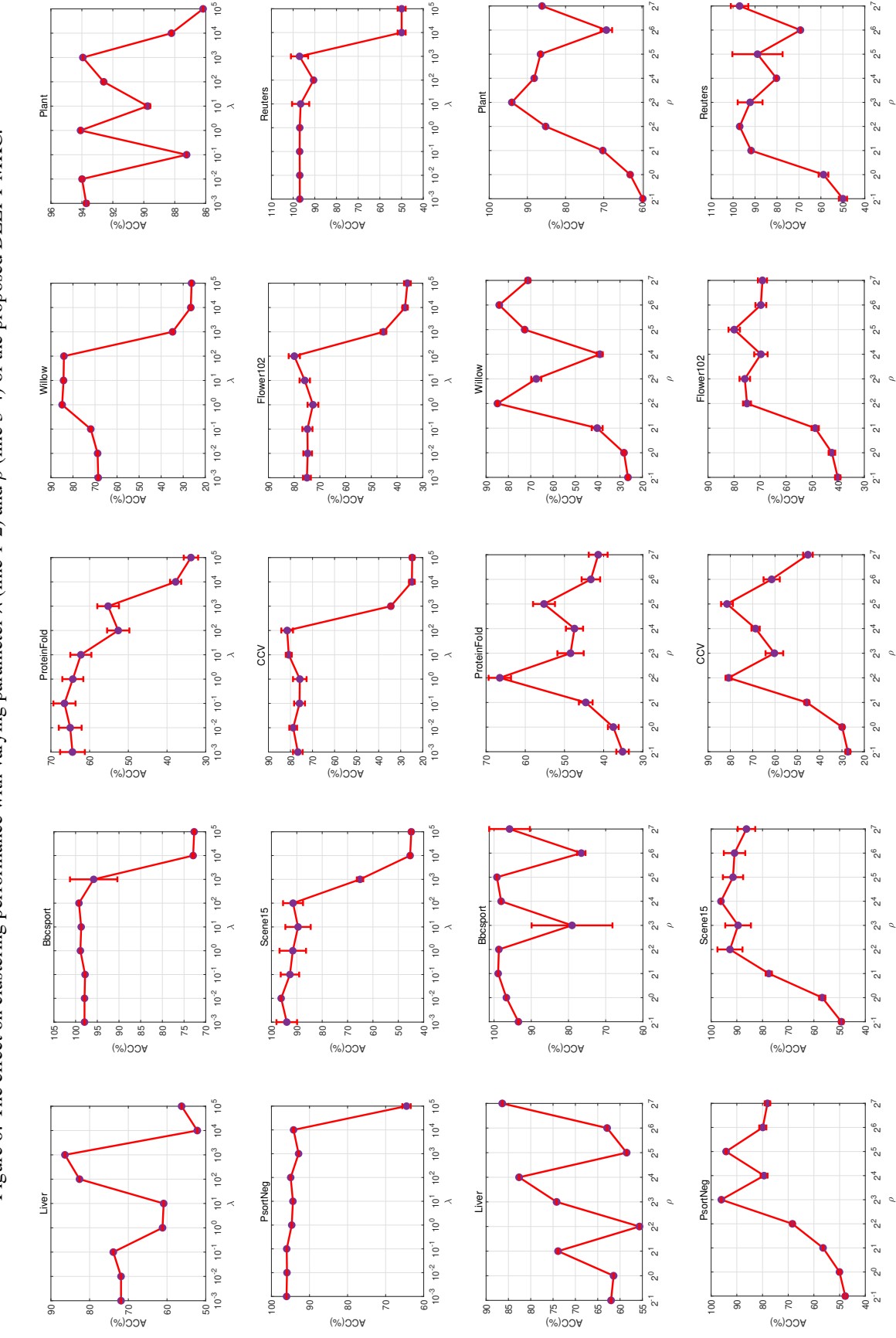

Figure 6: The effect on clustering performance with varying parameter $\lambda$ (line 1-2) and $\rho$ (line 3-4) of the proposed DLEFT-MKC.

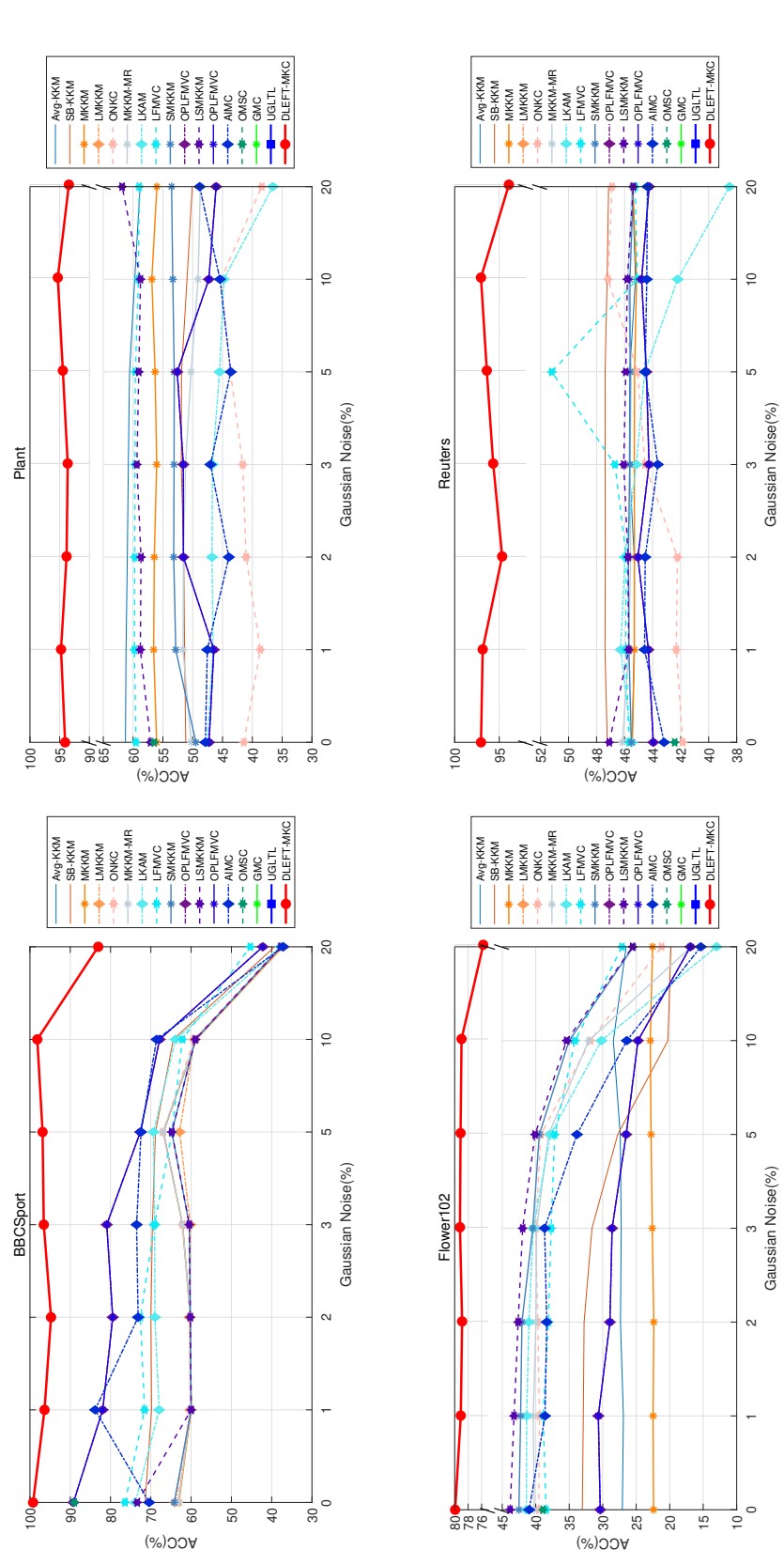

Figure 7: Comparison of comprehensive clustering performance fluctuations between DLEFT-MKC algorithm and comparative algorithm under different degrees of Gaussian noise (0, 1, 2, 3, 5, 10, 20%).

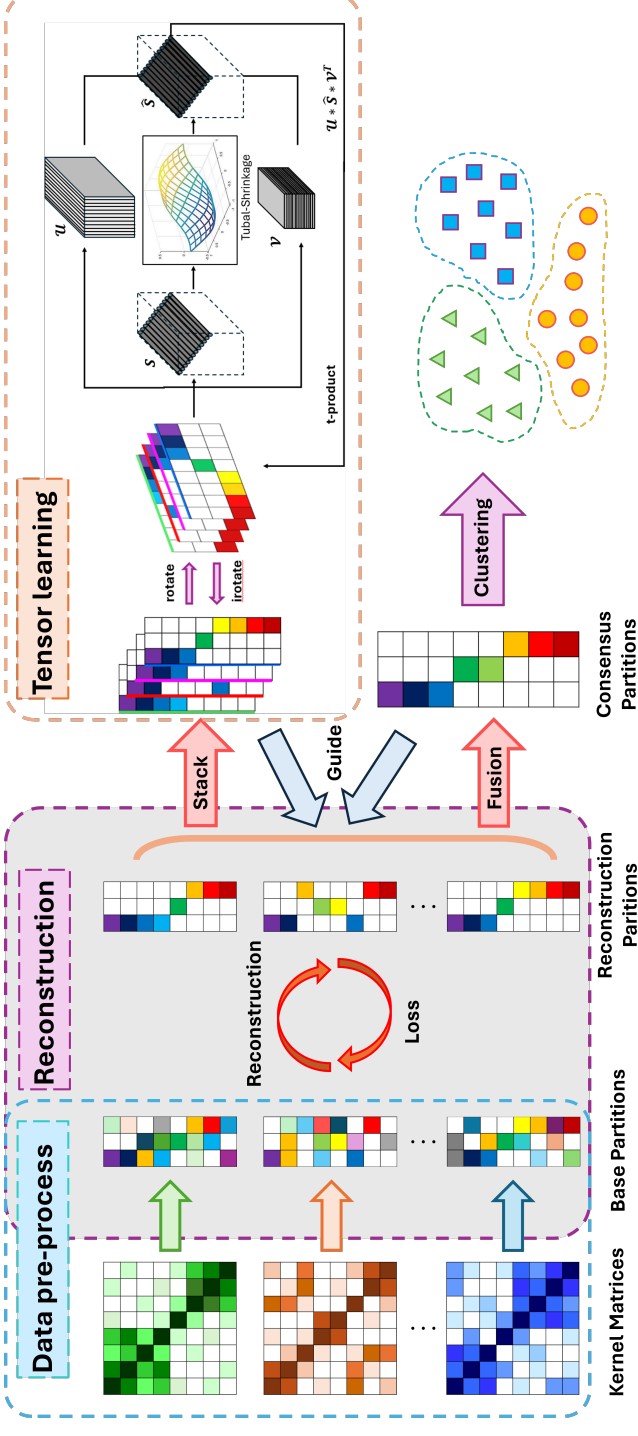

Figure 8: The framework diagram of the proposed DLEFT-MKC algorithm.

