# OpenReview forum: "DLEFT-MKC: Dynamic Late Fusion Multiple Kernel Clustering with Robust Tensor Learning via Min-Max Optimization"
_ICLR.cc/2025/Conference — ICLR 2025 Spotlight_

### Official Review · Reviewer_JfTa · 2024-11-01

**Soundness:** 3
**Presentation:** 3
**Contribution:** 2
**Rating:** 6
**Confidence:** 5

**Summary:**

This paper presents a novel algorithm, Dynamic Late Fusion Multiple Kernel Clustering with Robust Tensor Learning (DLEFT-MKC), which addresses key limitations in existing multiple kernel clustering methods. It dynamically optimizes base partition matrices, introduces min-max optimization for enhanced robustness and generalization, and reconstructs decision layers to capture high-order associations across views. These advancements improve computational efficiency and enable comprehensive extraction of multiple kernel information, leading to effective consensus clustering partitions.

**Strengths:**

1. The experimental results are comprehensive, covering datasets of varying scales, comparative methods, and diverse visualization experiments.

2. The overall writing is clear and easy to understand.

**Weaknesses:**

1. The motivation for this paper is unclear and lacks persuasiveness. For instance, the authors mention that one of the drawbacks of existing methods is their reliance on fixed initial base partitions, but they do not define what "initial base partitions" are. Additionally, while the authors highlight that current multiple kernel k-means methods often lack adaptive mechanisms for varying internal data distributions, the paper does not provide a solution to this issue.

2. A framework diagram of the proposed algorithm should be included in the manuscript to aid understanding.

3. The paper's novelty is insufficient, as it primarily integrates three related components from existing work, suggesting that it does not offer any new contributions.

4. The paper lacks a theoretical analysis of computational complexity.

5. Replacing Figure 4 with a table to present the comparison of runtime would provide a clearer and more intuitive representation.

6. The reference list contains inconsistencies, with some conference names abbreviated and others in full, and some entries including DOI information while others do not. These details should be made consistent.

**Questions:**

The authors claim that existing multiple kernel k-means methods often lack adaptive mechanisms for varying internal data distributions, stating that their proposed method addresses this issue. However, where is this adaptation reflected in the paper, and how does the method achieve it?

**Details Of Ethics Concerns:**

N.A.

---

> ### Author Response · Authors · 2024-11-20
> **--- Rebuttal by Authors 1/3 ---**
>
> Thank you so much for the recognition and valuable comments! We have revised the manuscript significantly. Also, the corresponding responses are as follows.
>
> **1. Definition, Motivation, and  Explanation**
>
> Thanks. *Definition of base partition*:
> We define the base partition as the matrices of n*k for each kernel view, as the Fp, obtained by directly performing eigenvalue decomposition on kernel matrices and obtaining the large k eigenvalue corresponding eigenvectors consisting of the matrices.
>
>
> *Adaptive mechanisms*: Adaptive mechanisms refer to the dynamic reconstruction of partition matrices and the min-max paradigm.
>
> - *Dynamic Reconstruction*: Base partition matrices can directly capture information from multiple kernels. However, this process has limitations. Specifically, the information captured is often limited (in amount or quality) and heavily dependent on the initialization. As a result, the captured information may not be sufficient; poor initialization can lead to suboptimal partitions, thereby causing performance bottlenecks.
>
> To address this issue, we have designed a dynamic reconstruction strategy that adaptively learns more optimal partitions. This approach enables the algorithm to continuously capture more beneficial information, thereby enhancing the subsequent clustering tasks. This is achieved through iterative refinement, where the algorithm continuously updates the base partitions based on the current state; and this continuous learning process ensures that the algorithm remains adaptive and responsive to the underlying data structure. By dynamically learning base partitions, we can guide the learning process more effectively and mitigate the negative impact of poor initialization.
>
> - *Min-max Paradigm*: The min-max paradigm introduced in our proposed algorithm is centered around the philosophical concept of "finding optimism in pessimism." Specifically, the algorithm aims to minimize the parameter $\gamma$ while simultaneously maximizing the overall objective function. This adversarial approach enhances the stability and robustness of the algorithm.
>
> In traditional optimization paradigms, if a particular view temporarily achieves better performance, the algorithm tends to increase the weight of that view, thereby diminishing the influence of other views. This can lead to the algorithm becoming overly reliant on a single kernel view, which may compromise its overall performance. In contrast, our proposed min-max paradigm avoids this issue by preventing the algorithm from overly favoring any single view. This ensures a balanced and stable performance across different views.
>
> To further validate the robustness of our proposed algorithm, we conducted additional experiments with noisy data. The visual comparisons of clustering results are presented in Fig.7 in the Appendix.
>
> The results show that our proposed algorithm consistently outperforms compared algorithms in terms of clustering performance and exhibits more excellent stability when subjected to noise interference. Unlike other algorithms, which show significant degradation in performance under noisy conditions, our proposed algorithm demonstrates minimal fluctuation, highlighting its superior robustness.
>
> Based on these positive effects, our proposed algorithm achieves more stable and reliable clustering results, especially in the presence of noise. As demonstrated in Fig.7, the proposed algorithm not only outperforms baseline algorithms in terms of clustering performance but also shows minimal fluctuation when subjected to noise interference, highlighting its superior robustness.
>
> **2. Framework diagram**
>
> Thanks. To aid understanding, we have added a framework diagram of the proposed DELFT-MKC, as shown in Fig.8.
>
>
> The proposed DLEFT-MKC algorithm represents a significant advancement in the field of multiple kernel clustering. By combining the late fusion paradigm, tensor techniques, and min-max optimization in a novel and integrated manner, we achieve substantial improvements in both computational efficiency and clustering performance. The dynamic reconstruction of base partitions and the new optimization strategy further enhance the algorithm's robustness and adaptability, making it a valuable tool for a wide range of applications.

---

> ### Author Response · Authors · 2024-11-20
> **--- Rebuttal by Authors 2/3 ---**
>
> **3. Novelty and Contributions of the Proposed DLEFT-MKC Algorithm**
>
> Although the proposed DLEFT-MKC algorithm leverages tensor techniques, late fusion paradigm, and min-max optimization, it is not a simple combination of existing methods but a novel and innovative approach.
>
> (1) **Late Fusion Paradigm in Tensor Techniques**:
>    - Existing algorithms apply tensor techniques to Multiple Kernel Clustering (MKC) by stacking kernel matrices or affinity matrices into an $ n \times n \times m $ tensor. However, DLEFT-MKC introduces the late fusion paradigm, which allows us to construct an $ n \times k \times m $ tensor. This significantly improves the computational efficiency of using tensor techniques by reducing the dimensionality and complexity of the tensor operations.
>
> (2) **Min-Max Optimization in Tensor-Based MKC**:
>    - For the first time, we incorporate min-max optimization into tensor-based MKC. This is a highly novel approach that enhances the robustness of the algorithm. By integrating the philosophical concept of "finding optimism in pessimism," the proposed algorithm ensures robust performance even when faced with less favorable datasets. This results in better clustering outcomes under various conditions.
>    - We propose a new optimization algorithm that combines Alternating Direction Method of Multipliers (ADMM) and Reduced Gradient Descent Method (RGDM). This hybrid approach effectively solve the resultant problem and provide optimization method references for the community.
>
> (3) **Dynamical Reconstruction and Calibration of Base Partition Matrices**:
>    - Another first-of-its-kind contribution is the dynamical reconstruction and calibration of base partition matrices. This innovation overcomes the representational bottleneck of traditional base partition matrices, leading to enhanced clustering performance. By dynamically learning the base partitions, the algorithm can better adapt to the underlying data structure, thereby improving the overall clustering quality.
>
> ### Overall Contributions
>
> - **Unified Framework**: The late fusion paradigm, tensor techniques, and min-max optimization have been individually explored in the literature. However, DLEFT-MKC is the first to organically integrate these components into a unified framework. This integration results in significant improvements in both computational efficiency and clustering performance.
> - **Dynamic Reconstruction and New Optimization Strategy**: We introduce dynamic reconstruction of base partition matrices and a novel optimization strategy that effectively addresses the challenges posed by our proposed problem. These contributions ensure that the algorithm remains robust and efficient, even in complex and diverse datasets.
>
> ### Conclusion
>
> The proposed DLEFT-MKC algorithm represents a significant advancement in the field of multiple kernel clustering. By combining the late fusion paradigm, tensor techniques, and min-max optimization in a novel and integrated manner, we achieve substantial improvements in both computational efficiency and clustering performance. The dynamic reconstruction of base partitions and the new optimization strategy further enhance the algorithm's robustness and adaptability, making it a valuable tool for a wide range of applications.

---

> ### Author Response · Authors · 2024-11-20
> **--- Rebuttal by Authors 3/3 ---**
>
> **4. Analysis of Computational Complexity**
>
> Thanks. It may be due to page limitations, therefore, we will include enough theoretical analysis of computational complexity in the final version with extra page space. We analyze the computational complexity as follows,
>
> The time complexity of the proposed DLEFT-MKC algorithm is $ \mathcal{O}(nmk \log(m) + nmk^2) $ per iteration, with the following detailed derivation:
>
> (1) **Updating $\hat{F}_p$ and $T_p$**:
>    - According to Eq.8 and Eq.16, these updates require $2m$ times economic SVD operations with a complexity of $ \mathcal{O}(nk^2) $.
>    - Therefore, the total complexity for these updates is $ \mathcal{O}(2m \cdot nk^2) = \mathcal{O}(2mnk^2) $.
>
> (2) **Updating $\boldsymbol{\beta}$ and $F^{*}$**:
>    - Based on Eq.11, these updates involve a reduced gradient descent method with a complexity of $ \mathcal{O}(nk^2 l) $, where $ l $ is the number of iterations, typically small.
>
> (3)  **Updating $\mathcal{A}$**:
>    -  Based on Eq.15, this update involves solving proximal operators, with a complexity of $ \mathcal{O}(nmk \log(m) + nmk^2) $.
>
> Given that $ n \gg k $ and $ n \gg m $, the overall time complexity of the algorithm is dominated by the most complex term, resulting in $ \mathcal{O}(nmk \log(m) + nmk^2) $.
>
> ### Comparison with Other Algorithms
>
> Furthermore, we list the time complexity of all compared methods in Tab.5 for a theoretical comparison of time complexity with other algorithms.
> As evident, the computational efficiency of the proposed DLEFT-MKC algorithm is linear, contrasting sharply with the cubic time complexity $ \mathcal{O}(n^3 mk) $ of common MKC algorithms. This aligns well with our efficiency goals.
>
> Our proposed algorithm is similar in complexity to other late fusion-based methods, such as OPLFMVC and LFMVC. OPLFMVC generally maintains the best efficiency due to its one-pass method.
> Yet, the proposed DLEFT-MKC algorithm not only achieves superior clustering performance but also maintains high computational efficiency. This makes it a valuable tool for handling large-scale and complex datasets, where both performance and efficiency are critical.
>
>
> **5. Running Time Comparison**
>
> Thanks. The table (Tab.10) comparing running time was included in the appendix of the submitted manuscript.
> Note that, due to *page limitations*, we had to take it into the appendix, and We had to adopt a way to visually compare and save space to demonstrate the comparison and discussion of computational efficiency.
>
> We will replace it in the final version with extra page space to provide a clearer and more intuitive representation.
>
> **6. Reference List Revision**
>
> Thank you for pointing out the issues with the references. We have carefully revised and standardized the references, including the full names and abbreviations of conferences and journals, capitalization, and other information.
>
> **Last**
>
> We are happy to any questions, suggestions or comments. Please do not hesitate to tell us. We will response timely.

---

> ### Author Response · Authors · 2024-11-25
> **The deadline of Discussion Period is approaching**
>
> Dear Reviewer JfTa,
>
> We highly appreciate your valuable and insightful reviews. We hope the above response has addressed your concerns. If you have any other suggestions or questions, feel free to discuss them. We are very willing to discuss them with you in this period.
>
> If your concerns have been addressed, would you please consider raising the score? It is very important for us and this research. Thanks again for your professional comments and valuable time!
>
> Best wishes,
>
> Authors

---

> ### Comment · Reviewer_JfTa · 2024-11-28
>
> Thank you for your response. Most of my concerns have been resolved.

---

> > ### Author Response · Authors · 2024-12-02
> > **Thanks for Reviewer JfTa!**
> >
> > Dear Reviewer JfTa,
> >
> > Thank you for your support and recognition of our contributions.
> >
> > We will make comprehensive revisions based on your valuable feedback in the final version to further improve and perfect the quality of the paper.
> >
> > We sincerely appreciate the time and effort you have invested in providing such profound and insightful suggestions.
> >
> > Best regards,
> >
> > Authors of Paper 1283

---

### Official Review · Reviewer_Noi6 · 2024-11-01

**Soundness:** 3
**Presentation:** 4
**Contribution:** 3
**Rating:** 8
**Confidence:** 4

**Summary:**

This paper proposes a novel DLEFT-MKC, which effectively overcomes the representational bottleneck of base partition matrices and facilitates the learning of meaningful high-order cross-view information. Specifically, it is the first to incorporate a min-max optimization paradigm into tensor-based MKC, enhancing algorithm robustness and generalization. Extensive experimental results across various benchmark datasets validate the superior effectiveness and efficiency of the proposed algorithm.

**Strengths:**

1. The paper is well-organized and written without technical errors.
2. The motivation behind the paper is clear. The proposed method is innovative as it is the first to incorporate a min-max optimization paradigm into tensor-based MKC and the first to dynamically reconstruct base partition matrices from late fusion multi-view clustering.
3. The experiments are comprehensive and solid, involving comparisons with over 20 competing algorithms. The proposed method consistently achieves the best performance, and the experiments have been evaluated from multiple perspectives.
4. The experimental results demonstrate a substantial improvement in clustering performance, which is particularly exciting when coupled with a significant increase in computational efficiency.

**Weaknesses:**

1. The notation is somewhat difficult to follow. Some symbols are used without explanation. A table summarizing the notation would be beneficial.
2. The case where $\gamma_p = 0$ and $\Delta \leq 0$ is missing from the discussion in Eq.(14).
3. There is insufficient discussion on the proposed algorithm. The objective expression appears complex, and the max-min-max form raises questions about how it ensures rapid convergence. While the experimental results show a notable improvement in computational efficiency compared to most other algorithms, there is a lack of analysis on computational complexity to explain this phenomenon.
4. The cluster assignments in Fig.2 are not easily distinguishable, which is insufficient to validate the effectiveness of the learning process. The authors should provide additional visualizations to substantiate this contribution. For example, on the Reuters dataset, most algorithms achieve an average accuracy around 45%, whereas the proposed algorithm achieves an accuracy of 97%. Such a high accuracy rate raises concerns, and the authors need to provide an explanation.
5. The author should provide the code and sufficient information for readers to reproduce the experimental results.

**Questions:**

As mentioned above.

---

> ### Author Response · Authors · 2024-11-20
> **--- Rebuttal by Authors 1/2 ---**
>
> Thank you so much for the recognition and valuable comments! We have revised the manuscript significantly. Also, the corresponding responses are as follows.
>
> **1. Notation Table**
>
> Thanks. To improve clarity and readability, we have included a notation table as shown in Tab.4, which provides a comprehensive list and explanation of the main symbols used in this article.
>
> **2. Descent Direction  Expression**
>
> when $\gamma_p=0$ and $\nabla\leq 0$, the descent direction is the normal form: $-\left[\bigtriangledown\mathcal{G}(\boldsymbol{\gamma})\right]_{p}$. We have fixed the equation in the manuscript. To ensure that all cases are covered, we have revised Eq.(14) in the manuscript.
>
> $ \quad \quad \ \ | \  0  \quad \quad \quad  \quad \quad \ \ $ if $\gamma_p = 0$ and $[\nabla\mathcal{G}(\boldsymbol{\gamma})]_p > 0$,
>
> $v_p =  \ |  - [\nabla\mathcal{G}(\boldsymbol{\gamma})]_{u}  \quad$   if $p=u$,
>
> $ \quad \quad \ \ |   - [\nabla\mathcal{G}(\boldsymbol{\gamma})]_{p}  \quad$ otherwise.
>
> **3. Computational Complexity**
>
> Thanks. Although the objective function of the proposed algorithm appears complex, we have designed an efficient and effective optimization strategy to solve it. The computational complexity of this strategy is $\mathcal{O}(nmk \log(m) + nmk^2)$ per iteration. Despite the complexity, the algorithm maintains high computational efficiency and achieves excellent clustering performance. The detailed derivation process is as follows:
>
> The time complexity of the proposed DLEFT-MKC algorithm is $ \mathcal{O}(nmk \log(m) + nmk^2) $ per iteration, with the following detailed derivation:
>
> (1) **Updating $\hat{F}_p$ and $T_p$**:
>    - According to Eq.8 and Eq.16, these updates require $2m$ times economic SVD operations with a complexity of $ \mathcal{O}(nk^2) $.
>    - Therefore, the total complexity for these updates is $ \mathcal{O}(2m \cdot nk^2) = \mathcal{O}(2mnk^2) $.
>
> (2) **Updating $\boldsymbol{\beta}$ and $F^{*}$**:
>    - Based on Eq.11, these updates involve a reduced gradient descent method with a complexity of $ \mathcal{O}(nk^2 l) $, where $ l $ is the number of iterations, typically small.
>
> (3)  **Updating $\mathcal{A}$**:
>    -  Based on Eq.15, this update involves solving proximal operators, with a complexity of $ \mathcal{O}(nmk \log(m) + nmk^2) $.
>
> Given that $ n \gg k $ and $ n \gg m $, the overall time complexity of the algorithm is dominated by the most complex term, resulting in $ \mathcal{O}(nmk \log(m) + nmk^2) $.
>
> ### Comparison with Other Algorithms
>
> Furthermore, we list the time complexity of all compared methods in Tab.5 for a theoretical comparison of time complexity with other algorithms.
> As evident, the computational efficiency of the proposed DLEFT-MKC algorithm is linear, contrasting sharply with the cubic time complexity $ \mathcal{O}(n^3 mk) $ of common MKC algorithms. This aligns well with our efficiency goals.
>
> Our proposed algorithm is similar in complexity to other late fusion-based methods, such as OPLFMVC and LFMVC. OPLFMVC generally maintains the best efficiency due to its one-pass method.
> Yet, the proposed DLEFT-MKC algorithm not only achieves superior clustering performance but also maintains high computational efficiency. This makes it a valuable tool for handling large-scale and complex datasets, where both performance and efficiency are critical.

---

> ### Author Response · Authors · 2024-11-20
> **--- Rebuttal by Authors 2/2 ---**
>
> **4. Explanation of Experimental Results**
>
> Thanks. To facilitate easier and clearer distinguishing, we normalized the partition matrices and updated the visual cluster structure, as shown in Fig.2. We observe that during the clustering process, the cluster structure becomes increasingly clear and distinguishable as the DLEFT algorithm iterates. This clearly demonstrates the effectiveness of the DLEFT algorithm's learning process. The well-defined cluster structure also explains why the algorithm achieves such high clustering accuracy.
>
> On the other hand, some deep clustering methods or classification algorithms can achieve around 94% accurancy, and even 99% purity on the Reuters dataset [Automatic classification of national health service feedback(2022), Classification of multi-labeled text articles with reuters dataset using SVM(2022), Analysis of the Influence of Preprocessing Techniques on Text Classification Accuracy: An Investigation with the Naive Bayes Model and the Reuters-21578 Dataset(2021), A set theory based similarity measure for text clustering and classification(2020), Efficient deep embedded subspace clustering(2022), Transformed K-means clusterin(2021)]. To further evaluate the performance of GMC, LTBPL, UGLBL, and WTNNM on the Reuters dataset, we try to conduct tests on a computer with larger memory (since the 64GB memory was insufficient for these algorithms to run on the Reuters dataset). And achieve the following results:
> - GMC: 40% ACC, 19% PUR, 48% PUR, 15% RI, with 1657.2s running time.
> - UGLBL: 79% ACC, 70% PUR, 84% PUR, 66% RI, with 17095s running time.
> - LTBPL: We were unable to obtain results on a computer with 128GB of memory due to out-of-memory errors.
> - WTNNM: We were unable to obtain the results within the 8 days of running the algorithms (updated: 23 Nov), and we will update the results as soon as they are completed.
>
> *Note: By comparing these tensor-based clustering algorithms, particularly LTBPL and WTNNM, the efficiency of the proposed algorithm in this paper can be significantly highlighted.*
>
> These results indicate that most algorithms achieve only around 45% accuracy on the Reuters dataset primarily due to their inability to fully extract information from the complex data structure and learn an ideal data representation. In contrast, our proposed algorithm demonstrates very encouraging clustering performance on this dataset. The evolving clustering performance and visualized cluster structures (see Fig.1 and Fig.2) further confirm this.
>
> **5. Source Code and Reproductivity**
>
> Thanks. We upload the source codes to Anonymous GitHub: https://anonymous.4open.science/r/DLEFT-MKC-3F8F/.
> All experimental information is provided, including all utilized functions and selected parameters.
>
> **Last**
>
> We are happy to any questions, suggestions or comments. Please do not hesitate to tell us. We will response timely.

---

> ### Author Response · Authors · 2024-11-25
> **The deadline of Discussion Period is approaching**
>
> Dear Reviewer Noi6,
>
> We highly appreciate your valuable and insightful reviews. We hope the above response has addressed your concerns. If you have any other suggestions or questions, feel free to discuss them. We are very willing to discuss them with you in this period.
>
> Thank you so much for the recognition, professional comments, and valuable time! It is very important for us and this research.
>
> Best wishes,
>
> Authors

---

> ### Author Response · Authors · 2024-12-02
> **Thanks for Reviewer Noi6!**
>
> Dear Reviewer Noi6,
>
> Thank you for your support and recognition of our contributions.
>
> We will make comprehensive revisions based on your valuable feedback in the final version to further improve and perfect the quality of the paper.
>
> We sincerely appreciate the time and effort you have invested in providing such profound and insightful suggestions.
>
> Best regards,
>
> Authors of Paper 1283

---

> > ### Comment · Reviewer_Noi6 · 2024-12-03
> >
> > Thank you for your response. My concerns have been addressed, so I recommend accepting this manuscript.

---

### Official Review · Reviewer_xS8p · 2024-11-01

**Soundness:** 3
**Presentation:** 3
**Contribution:** 3
**Rating:** 8
**Confidence:** 3

**Summary:**

The authors present a novel method for multiple kernel clustering called DLEFT-MKC, which addresses the problem of representational bottleneck of base partition matrices. This paper proposes a min-max optimization paradigm in tensor-based MKC to improve both performance and robustness. In addition, the authors use tensor learning to capture the high-order correlations between kernels. Finally, the effectiveness of DLEFT-MKC is demonstrated through extensive experiments on benchmark datasets.

**Strengths:**

1) This study is the first to apply a min-max optimization paradigm to tensor-based multi-kernel clustering, pioneering a new approach to improve performance and robustness in clustering.
2）The overall structure of the paper is complete and well-organized.
3) The formulas in the paper are written correctly and the derivations are correct.
4) Another bright point in this paper is that it combines the RGDE with ADMM to optimize their problem, this combination is a well-established technique to solve convex optimization problems.

**Weaknesses:**

1) The complexity of the proposed method is not analyzed.
2) The base partition matrices directly capture data information from multiple kernels. Why reconstruct the dynamic partition matrices? It would be better if the authors clearly introduced this design.
3) Why are the standard deviation (std) values for your method so high on the Reuters and CCV datasets in Table 2?
4) As far as I know, datasets like Caltech101-7, Caltech101-20, and Caltech101-all are the most commonly used for multiple kernel clustering, and it would be more convincing to compare the experimental results on these datasets. Please show a comparative experiment of the proposed method on at least one of them.

**Questions:**

See the weakness.

---

> ### Author Response · Authors · 2024-11-20
> **--- Rebuttal by Authors 1/2 ---**
>
> Thank you so much for the recognition and valuable comments! We have revised the manuscript significantly. Also, the corresponding responses are as follows.
>
> **1. Computational Complexity of the Proposed DLEFT-MKC Algorithm**
>
> Thanks. The time complexity of the proposed DLEFT-MKC algorithm is $ \mathcal{O}(nmk \log(m) + nmk^2) $ per iteration, with the following detailed derivation:
>
> (1) **Updating $\hat{F}_p$ and $T_p$**:
>    - According to Eq.8 and Eq.16, these updates require $2m$ times economic SVD operations with a complexity of $ \mathcal{O}(nk^2) $.
>    - Therefore, the total complexity for these updates is $ \mathcal{O}(2m \cdot nk^2) = \mathcal{O}(2mnk^2) $.
>
> (2) **Updating $\boldsymbol{\beta}$ and $F^{*}$**:
>    - Based on Eq.11, these updates involve a reduced gradient descent method with a complexity of $ \mathcal{O}(nk^2 l) $, where $ l $ is the number of iterations, typically small.
>
> (3)  **Updating $\mathcal{A}$**:
>    -  Based on Eq.15, this update involves solving proximal operators, with a complexity of $ \mathcal{O}(nmk \log(m) + nmk^2) $.
>
> Given that $ n \gg k $ and $ n \gg m $, the overall time complexity of the algorithm is dominated by the most complex term, resulting in $ \mathcal{O}(nmk \log(m) + nmk^2) $.
>
> ### Comparison with Other Algorithms
>
> Furthermore, we list the time complexity of all compared methods in Tab.5 for a theoretical comparison of time complexity with other algorithms.
> As evident, the computational efficiency of the proposed DLEFT-MKC algorithm is linear, contrasting sharply with the cubic time complexity $ \mathcal{O}(n^3 mk) $ of common MKC algorithms. This aligns well with our efficiency goals.
>
> Our proposed algorithm is similar in complexity to other late fusion-based methods, such as OPLFMVC and LFMVC. OPLFMVC generally maintains the best efficiency due to its one-pass method.
> Yet, the proposed DLEFT-MKC algorithm not only achieves superior clustering performance but also maintains high computational efficiency. This makes it a valuable tool for handling large-scale and complex datasets, where both performance and efficiency are critical.
>
> **2. Dynamic Reconstruction of Base Partition Matrices**
>
> Thanks. Base partition matrices can directly capture information from multiple kernels. However, this process has limitations. Specifically, the information captured is often limited (in amount or quality) and heavily dependent on the initialization. As a result, the captured information may not be sufficient; poor initialization can lead to suboptimal partitions, thereby causing performance bottlenecks.
>
> To address this issue, we have designed a dynamic reconstruction strategy that adaptively learns more optimal partitions. This approach enables the algorithm to continuously capture more beneficial information, thereby enhancing the subsequent clustering tasks. This is achieved through iterative refinement, where the algorithm continuously updates the base partitions based on the current state; and this continuous learning process ensures that the algorithm remains adaptive and responsive to the underlying data structure. By dynamically learning base partitions, we can guide the learning process more effectively and mitigate the negative impact of poor initialization.
>
> **3. Explanation of Experimental Results**
>
> Thanks. From Tab.2, it can be observed that the standard deviation (std) of the proposed algorithm is relatively high on the Reuters and CCV datasets. This indicates that in the learned clustering structures, there are some sample instances that exhibit ambiguity, meaning they have similar distances to multiple clusters. Due to the inherent randomness of the k-means algorithm, these ambiguous samples may be assigned to different clusters across multiple runs, leading to higher variance.
>
> Despite this, the high standard deviation is not necessarily a drawback, especially given the high clustering accuracy achieved by the proposed algorithm. In practice, maintaining a high clustering accuracy while allowing for higher variance can actually highlight samples that are difficult to classify or ambiguous. This can serve as a useful indicator for researchers to re-examine and verify these instances, potentially leading to further improvements in real-world clustering applications.
>
> Moreover, even in the worst-case scenario (when the std is at its lower bound), the proposed algorithm's clustering performance still significantly outperforms other algorithms. This underscores the robustness and effectiveness of the proposed DLEFT-MKC.
>
> Overall, the high standard deviation observed in the proposed algorithm on the Reuters and CCV datasets is not a significant drawback. It reflects the presence of ambiguous samples and the algorithm's ability to highlight them. This can aid researchers in verifying and improving the clustering results. Moreover, the algorithm's strong performance, even in the worst-case scenario, further validates its effectiveness and robustness.

---

> ### Author Response · Authors · 2024-11-20
> **--- Rebuttal by Authors 2/2 ---**
>
> **4. More Comparative Experiment**
>
> Thanks. We conduct additional experiments on Caltech datasets to evaluate the clustering performance of compared algorithms, where Caltech101-30T (3060 samples, 48 views, 102 clusters) denotes that each cluster has 30 samples. The results are shown below, and we will revise the manuscript and include them in the final version according to the page space.
>
>
> | Algorithm     | Caltech101-7 | Caltech101-20 | caltech101_30T | Caltech101-all |
> |--------------|--------------|---------------|----------------|----------------|
> | Avg-KKM      | 59.2         | 36.2          | 28.6           | 22.5           |
> | SB-KKM       | 64.3         | 37.8          | 33.9           | 24.7           |
> | MKKM         | 52.2         | 29.5          | 16.6           | 14.8           |
> | LMKKM        | 53.9         | 28.7          | -              | -              |
> | ONKC         | 69.4         | 40.4          | 28.2           | 25.8           |
> | MKKM-MR      | 68.9         | 40.3          | 30.6           | 25.7           |
> | LKAM         | 70.4         | 40.8          | 24.8           | 24.3           |
> | LFMVC        | 71.4         | 40.6          | 31.7           | 24.3           |
> | NKSS         | 64.3         | 37.8          | 33.9           | 24.7           |
> | SPMKC        | 64.4         | 43.4          | 22.1           | -              |
> | HMKC         | 74.6         | 44.4          | 35.3           | -              |
> | SMKKM        | 68.2         | 40.1          | 30.5           | 23.6           |
> | AIMC         | 73.5         | 44.2          | 29.5           | 31.4           |
> | OMSC          | 69.0         | 39.9          | 30.9           | 23.7           |
> | OPLFMVC      | 59.9         | 43.8          | 27.2           | 29.9           |
> | LSMKKM       | 75.5         | 44.1          | 28.9           | 28.6           |
> | HFLSMKKM     | 53.2         | 32.8          | 25.1           | -              |
> | GMC          | 68.7         | 50.8          | 24.4           | 24.4           |
> | LTBPL        | 79.1         | 56.9          | -              | -              |
> | UGLTL        | 78.9         | 42.6          | 87.6           | 53.0           |
> | WTNNM        | 82.0         | 59.6          | -              | -             |
> | KCGT         | 74.9         | 52.3          | 86.6           | -              |
> |DLEFT-MKC    | 83.5         | 57.3          | 89.0           | 59.3            |
>
> Note: Some results were not reported due to out-of-memory errors or because the computations had not been completed within the allotted time (exceeding 7 days)
>
> **Last**
>
> We are happy to any questions, suggestions or comments. Please do not hesitate to tell us. We will response timely.

---

> ### Author Response · Authors · 2024-11-25
> **The deadline of Discussion Period is approaching**
>
> Dear Reviewer xS8p,
>
> We highly appreciate your valuable and insightful reviews. We hope the above response has addressed your concerns. If you have any other suggestions or questions, feel free to discuss them. We are very willing to discuss them with you in this period.
>
> If your concerns have been addressed, would you please consider raising the score? It is very important for us and this research. Thanks again for your professional comments and valuable time!
>
> Best wishes,
>
> Authors

---

> > ### Comment · Reviewer_xS8p · 2024-11-29
> >
> > Thanks for the response. Most of my concerns have been fixed.

---

> > > ### Author Response · Authors · 2024-12-02
> > > **Thanks for Reviewer xS8p!**
> > >
> > > Dear Reviewer xS8p,
> > >
> > > Thank you for your support and recognition of our contributions.
> > >
> > > We will make comprehensive revisions based on your valuable feedback in the final version to further improve and perfect the quality of the paper.
> > >
> > > We sincerely appreciate the time and effort you have invested in providing such profound and insightful suggestions.
> > >
> > > Best regards,
> > >
> > > Authors of Paper 1283

---

### Official Review · Reviewer_UZb7 · 2024-11-04

**Soundness:** 3
**Presentation:** 4
**Contribution:** 3
**Rating:** 6
**Confidence:** 4

**Summary:**

The paper presents a dynamic late fusion multiple kernel clustering with robust tensor learning via min-max optimization, named DLEFT-MKC. The proposed method aims to address the robustness and generalization problem via a min-max objective. It utilizes t-TNN constraints to explore the high-order relationships and cluster structures between views. The unstable performance caused by the base partition matrices is mitigated by the dynamically adjusted partition matrices. Additionally, late fusion is introduced to reduce its computational complexity. These modules help enhance the clustering performance. The authors also conduct extensive experiments to validate the effectiveness and efficiency of DLEFT-MKC in terms of performance and time cost.

**Strengths:**

1.	The paper is well-written, and the motivations are clearly described;
2.	The idea that incorporates a min-max optimization paradigm into tensor-based MKC to enhance both performance and robustness in clustering tasks is novel to a certain degree.
3.	The paper conducts extensive experiments, and the results are adequately and logically discussed.

**Weaknesses:**

1.	For some justifications, more details should be provided.
2.	Leveraging t-SVD for tensor-based module optimization may recur high complexity, which is not consistent with the efficiency goal.
3.	The work seems to combine Wang’s work about late fusion with tensor learning. Leveraging tensor learning for multi-view clustering result fusion is not a novel idea, which has been utilized in many previous works. The main contribution of the work lies in the min-max optimization, which has also adopted by some previous works. Therefore, the novelty of the work might not be so convincing.

**Questions:**

1.	What does “robustness and generalization” refer to in this work? Does it mean less sensitivity to parameters?
2.	The underlying reason explaining the min-max optimization’s positive effect on robustness should be better explained.
3.	Eq. (4), $\gamma_p$ is used to balance the effect of each view, while in Eq. (6), $\gamma_p^2$ is adopted. We notice that the constraints on $\gamma_p$ in Eq. (4) and Eq. (6) remain the same, but why Eq. (6) adopts $\gamma_p^2$ instead?
4.	Though DLEFT-MKC is a late fusion approach, it is based on the third-order tensor and updating tensor requires t-SVD which is time-consuming. So why it has lower running on several datasets compared with other late fusion-based methods like OPLFMVC and LFMVC.

---

> ### Author Response · Authors · 2024-11-20
> **--- Rebuttal by Authors 1/3 ---**
>
> Thank you so much for the recognition and valuable comments! We have revised the manuscript significantly. Also, the corresponding responses are as follows.
>
> **1. Robustness and Generalization**
>
> Thanks. *Robustness* refers to an algorithm's ability to maintain its clustering performance in adverse conditions, specifically its capability to withstand pessimistic scenarios and noise interference. This robustness is a direct result of the *min-max paradigm* introduced in our proposed algorithm, which embodies the philosophical concept of "searching optimism in pessimism."
> *Generalization* refers to the algorithm's ability to achieve excellent clustering performance across a wide range of datasets.
>
> **Additional Noise Experiments**: To further validate the robustness of our proposed algorithm, we conducted additional experiments with noisy data. The visual comparisons of clustering results are presented in Fig.7 in the Appendix.
>
> The results show that our proposed algorithm consistently outperforms compared algorithms in terms of clustering performance and exhibits more excellent stability when subjected to noise interference. Unlike other algorithms, which show significant degradation in performance under noisy conditions, our proposed algorithm demonstrates minimal fluctuation, highlighting its superior robustness.
>
> **2. Min-Max Paradigm and Its Benefits**
>
> Thanks. The *min-max paradigm* introduced in our proposed algorithm is centered around the philosophical concept of "finding optimism in pessimism." Specifically, the algorithm aims to minimize the parameter $\gamma$ while simultaneously maximizing the overall objective function. This adversarial approach enhances the stability and robustness of the algorithm.
>
> In traditional optimization paradigms, if a particular view temporarily achieves better performance, the algorithm tends to increase the weight of that view, thereby diminishing the influence of other views. This can lead to the algorithm becoming overly reliant on a single kernel view, which may compromise its overall performance. In contrast, our proposed min-max paradigm avoids this issue by preventing the algorithm from overly favoring any single view. This ensures a balanced and stable performance across different views.
>
> Based on these positive effects, our proposed algorithm achieves more stable and reliable clustering results, especially in the presence of noise. As demonstrated in Fig.7, the proposed algorithm not only outperforms baseline algorithms in terms of clustering performance but also shows minimal fluctuation when subjected to noise interference, highlighting its superior robustness.
>
> **3. Constraints on $\gamma_p$**
>
> Thanks. The constraints on $\gamma_p$ in Eq.(4) and Eq.(6) differ. Specifically, in Eq.(4), the constraint $\gamma \in \nabla$ denotes that the sum of $\gamma_p^2$ is 1, i.e., $\sum_{p=1}^m \gamma_p^2 = 1$. In contrast, in Eq.(6), the constraint $\gamma \in \Delta$ denotes that the sum of $\gamma_p$ is 1, i.e., $\sum_{p=1}^m \gamma_p = 1$.
>
> - **Eq.(4)**: $\gamma \in \nabla$ implies $\sum_{p=1}^m \gamma_p^2 = 1$, which is a quadratic constraint. When $\gamma$ is subject to a quadratic constraint, we use $\gamma_p$ directly.
> - **Eq.(6)**: $\gamma \in \Delta$ implies $\sum_{p=1}^m \gamma_p = 1$, which is a linear constraint. When $\gamma$ is subject to a linear constraint, we use $\gamma_p^2$ instead.
>
> In this way, the constraint ensures that the weights are distributed more evenly across different views, preventing the trivial solution: any single view from dominating the others, i.e., one $\gamma_p$ is 1 and the others are 0.
>
> In addition, we believe that the algorithm would maintain its good performance if the constraint $\gamma \in \Delta$ (linear constraint) is adjusted to $\gamma_p^2 \in \nabla$ (quadratic constraint). The linear constraint $\gamma \in \Delta$ is more intuitive and easier to interpret, as it directly represents the weights of different views summing to one.

---

> ### Author Response · Authors · 2024-11-20
> **--- Rebuttal by Authors 2/3 ---**
>
> **4. Computational Efficiency of the Proposed DLEFT-MKC Algorithm**
>
> Although computing the full t-SVD for an $ n_1 \times n_2 \times n_3 $ tensor incurs a significant time cost, with a complexity of $ \mathcal{O}(n_1 n_2 n_3^2 + n_1^3 n_3 + n_2^3 n_3 + n_1 n_2 \min(n_1, n_2) n_3) $, the proposed DLEFT-MKC algorithm does not require a complete t-SVD during the optimization process. Instead, it only needs to compute partial t-SVDs when updating the tensor $\pmb{\mathcal{A}}$. Specifically, updating $\pmb{\mathcal{A}}$ in Eq.15 involves solving proximal operators, with a complexity of $ \mathcal{O}(nmk \log(m) + nmk^2) $.
>
> The time complexity of the proposed DLEFT-MKC algorithm is $ \mathcal{O}(nmk \log(m) + nmk^2) $ per iteration, with the following detailed derivation:
>
> (1) **Updating $\hat{F}_p$ and $T_p$**:
>    - According to Eq.8 and Eq.16, these updates require $2m$ times economic SVD operations with a complexity of $ \mathcal{O}(nk^2) $.
>    - Therefore, the total complexity for these updates is $ \mathcal{O}(2m \cdot nk^2) = \mathcal{O}(2mnk^2) $.
>
> (2) **Updating $\boldsymbol{\beta}$ and $F^{*}$**:
>    - Based on Eq.11, these updates involve a reduced gradient descent method with a complexity of $ \mathcal{O}(nk^2 l) $, where $ l $ is the number of iterations, typically small.
>    - Therefore, the total complexity for these updates is $ \mathcal{O}(nk^2 l) $.
>
> (3)  **Updating $\pmb{\mathcal{A}}$ in Eq.15**:
>    - This update requires a complexity of $ \mathcal{O}(nmk \log(m) + nmk^2) $.
>
> Given that $ n \gg k $ and $ n \gg m $, the overall time complexity of the algorithm is dominated by the most complex term, resulting in $ \mathcal{O}(nmk \log(m) + nmk^2) $.
>
> ### Comparison with Other Algorithms
>
> Furthermore, we list the time complexity of all compared methods in Tab.5 for a theoretical comparison of time complexity with other algorithms.
> As evident, the computational efficiency of the proposed DLEFT-MKC algorithm is linear, contrasting sharply with the cubic time complexity $ \mathcal{O}(n^3 mk) $ of common MKC algorithms. This aligns well with our efficiency goals.
>
> Our proposed algorithm is similar in complexity to other late fusion-based methods, such as OPLFMVC and LFMVC. On several small datasets, factors such as the number of iterations, specific computational mechanisms, and CPU usage can influence running time. OPLFMVC generally maintains the best efficiency due to its one-pass method.
> Yet, the proposed DLEFT-MKC algorithm not only achieves superior clustering performance but also maintains high computational efficiency. This makes it a valuable tool for handling large-scale and complex datasets, where both performance and efficiency are critical.

---

> ### Author Response · Authors · 2024-11-20
> **--- Rebuttal by Authors 3/3 ---**
>
> **5. Novelty and Contributions of the Proposed DLEFT-MKC Algorithm**
>
> Although the proposed DLEFT-MKC algorithm leverages tensor techniques, late fusion paradigm, and min-max optimization, it is not a simple combination of existing methods but a novel and innovative approach.
>
> (1) **Late Fusion Paradigm in Tensor Techniques**:
>    - Existing algorithms apply tensor techniques to Multiple Kernel Clustering (MKC) by stacking kernel matrices or affinity matrices into an $ n \times n \times m $ tensor. However, DLEFT-MKC introduces the late fusion paradigm, which allows us to construct an $ n \times k \times m $ tensor. This significantly improves the computational efficiency of using tensor techniques by reducing the dimensionality and complexity of the tensor operations.
>
> (2) **Min-Max Optimization in Tensor-Based MKC**:
>    - For the first time, we incorporate min-max optimization into tensor-based MKC. This is a highly novel approach that enhances the robustness of the algorithm. By integrating the philosophical concept of "finding optimism in pessimism," the proposed algorithm ensures robust performance even when faced with less favorable datasets. This results in better clustering outcomes under various conditions.
>    - We propose a new optimization algorithm that combines Alternating Direction Method of Multipliers (ADMM) and Reduced Gradient Descent Method (RGDM). This hybrid approach effectively solve the resultant problem and provide optimization method references for the community.
>
> (3) **Dynamical Reconstruction and Calibration of Base Partition Matrices**:
>    - Another first-of-its-kind contribution is the dynamical reconstruction and calibration of base partition matrices. This innovation overcomes the representational bottleneck of traditional base partition matrices, leading to enhanced clustering performance. By dynamically learning the base partitions, the algorithm can better adapt to the underlying data structure, thereby improving the overall clustering quality.
>
> ### Overall Contributions
>
> - **Unified Framework**: The late fusion paradigm, tensor techniques, and min-max optimization have been individually explored in the literature. However, DLEFT-MKC is the first to organically integrate these components into a unified framework. This integration results in significant improvements in both computational efficiency and clustering performance.
> - **Dynamic Reconstruction and New Optimization Strategy**: We introduce dynamic reconstruction of base partition matrices and a novel optimization strategy that effectively addresses the challenges posed by our proposed problem. These contributions ensure that the algorithm remains robust and efficient, even in complex and diverse datasets.
>
> ### Conclusion
>
> The proposed DLEFT-MKC algorithm represents a significant advancement in the field of multiple kernel clustering. By combining the late fusion paradigm, tensor techniques, and min-max optimization in a novel and integrated manner, we achieve substantial improvements in both computational efficiency and clustering performance. The dynamic reconstruction of base partitions and the new optimization strategy further enhance the algorithm's robustness and adaptability, making it a valuable tool for a wide range of applications.
>
> **Last**
>
> We are happy to any questions, suggestions or comments. Please do not hesitate to tell us. We will response timely.

---

> > ### Comment · Reviewer_UZb7 · 2024-11-28
> >
> > The authors' response has addressed some of my concerns.

---

> > > ### Author Response · Authors · 2024-12-02
> > > **Thanks for Reviewer UZb7!**
> > >
> > > Dear Reviewer UZb7,
> > >
> > > Thank you for your support and recognition of our contributions.
> > >
> > > We will make comprehensive revisions based on your valuable feedback in the final version to further improve and perfect the quality of the paper.
> > >
> > > We sincerely appreciate the time and effort you have invested in providing such profound and insightful suggestions.
> > >
> > > Best regards,
> > >
> > > Authors of Paper 1283

---

> ### Author Response · Authors · 2024-11-25
> **The deadline of Discussion Period is approaching**
>
> Dear Reviewer UZb7,
>
> We highly appreciate your valuable and insightful reviews. We hope the above response has addressed your concerns. If you have any other suggestions or questions, feel free to discuss them. We are very willing to discuss them with you in this period.
>
> If your concerns have been addressed, would you please consider raising the score? It is very important for us and this research. Thanks again for your professional comments and valuable time!
>
> Best wishes,
>
> Authors

---

### Author Response · Authors · 2024-11-20
**Rebuttal by Authors**

Thanks so much for the recognition and valuable comments of all reviewers! We have revised the manuscript significantly, and we conclude some aspects with which reviewers are concerned:

**1.The  Analysis of Computational Complexity**

**2. The Clearer Explanation and Motivation**

**3. The Novelty and Main Contributions**

**4. Additional Noise Experiments, Visualized in Fig.7**

**5. Extra Experiments on Caltech101 Datasets in Official Comments**

**6. Adding A Notation Table in Tab.4**

**7. Adding A Complexity Comparison Table in Tab.5**

**8. Updating Clustering Partition Figure in Fig.2**

**9. Adding A Framework Diagram Figure in Fig.8**

**10. Providing the Source Code and Settings for Reproductivity in Anonymous GitHub**  https://anonymous.4open.science/r/DLEFT-MKC-3F8F/

**11. Revising the Reference and Typos**

The tables and figures are updated in the revised submission PDF. Also, we will further improve the overall structure of the paper in the final version with additional page space.

We are happy to answer any questions, suggestions, or comments. Please do not hesitate to tell us. We will respond promptly.

---

### Meta-Review · Area_Chair_vYmK · 2024-12-19

**Metareview:**

The paper proposes DLEFT-MKC, a novel algorithm for multiple kernel clustering that overcomes limitations like fixed partition matrices and lack of high-order correlations. It uses min-max optimization and tensor learning to enhance representation and achieve better clustering performance with high efficiency. The overall review of the paper is positive, so the paper is recommended for acceptance at this time.

**Additional Comments On Reviewer Discussion:**

During the rebuttal period, the reviewers' opinions remained unchanged.

---

### Decision · Program_Chairs · 2025-01-22

Accept (Spotlight)